# Chalcopyrite Flotation, Molecular Design and Smart Industry: A Review

**DOI:** 10.3390/ijms26083613

**Published:** 2025-04-11

**Authors:** Luis A. Rios, Melanny J. Barraza, Pedro A. Robles, Gonzalo R. Quezada

**Affiliations:** 1Doctorado en Industria Inteligente, Facultad de Ingeniería, Pontificia Universidad Católica de Valparaíso, Valparaíso 2340000, Chile; lriosc3108@gmail.com; 2Escuela de Ingeniería Química, Pontificia Universidad Católica de Valparaíso, Valparaíso 2340000, Chile; melanny.barraza.t@mail.pucv.cl; 3Escuela de Ingeniería Química, Universidad del Bio-Bio, Concepción 4030000, Chile; grquezada@ubiobio.cl

**Keywords:** chalcopyrite, flotation, molecular design, machine learning

## Abstract

Chalcopyrite, the main source of copper worldwide, faces challenges in its flotation due to the complexity of its surface interactions and its coexistence with other minerals. Given the need for papers that show the current state of knowledge and new trends, this article reviews the developments of chalcopyrite flotation, with a focus on molecular design. A comprehensive bibliography search was conducted using keywords and specific queries in the Scopus database, applying inclusion and exclusion criteria to select the most relevant articles. The results were structured in three research periods, according to temporal and thematic criteria. The first period approaches the fundamentals of the process, considering variables as reagent dosage, surface chemistry and the influence of metal ions on recovery and selectivity. The second period explores the analysis and measurement techniques for the development of more selective and sustainable reagents. The third period analyzes the integration of advanced tools, such as molecular dynamic simulations and machine learning, into the understanding of adsorption mechanisms and custom reagent design. It is expected that this work will become a theoretical reference in future research and for mining companies that intend to innovate in their copper flotation and recovery processes.

## 1. Introduction

Currently, copper is a non-ferrous metal of high economic interest due to its properties like electrical conductivity, thermic conductivity, corrosion resistance and ductility, among other mechanical and structural properties [1,2]. It is mainly used in electrical cables, piping and heating, automobiles, air conditioning, telecommunications and industrial equipment, as well as others electrical and electronic products required in the construction, transportation, industrial, consumer and general goods sectors [1,2,3].

Additionally, it is important to note that copper is gaining greater relevance in the context of sustainable development due to its role in the energy transition toward renewable sources [4]. The increasing deployment of photovoltaic systems and wind farms has driven up copper demand, as these technologies require numerous electrical components, wiring systems, grounding systems, transformers and other equipment related to electricity transmission and distribution [4,5]. Future copper demand is expected to continue rising, exerting pressure on the global supply of this resource [6].

Worldwide, copper resources, mostly composed of oxide minerals and sulfide minerals, are estimated to be approximately 5.6 billion metric tons, with Chile, Australia and Peru the countries with the largest reserves [3]. Copper extraction from oxide minerals, such as malachite, cuprite and chrysocolla, requires hydrometallurgical processes, which include leaching, solvent extraction and electrowinning [7,8]. In contrast, sulfide minerals, such as chalcopyrite, bornite, chalcocite and covelline, are processed using techniques such as flotation, smelting, converting and electrorefining [9].

Of the minerals mentioned above, chalcopyrite is the most important in an economic and industrial view. Although it has theoretical copper content of 34.6%, lower than other minerals such as chalcocite or cuprite [10], its abundance in nature, the existence of large deposits that allow its extraction on a large scale, its compatibility with well-established metallurgical processes and its frequent association with valuable byproducts like gold, silver and molybdenum position it as the main source of copper worldwide [11,12,13,14].

Froth flotation, the most used process for chalcopyrite recovery, takes advantage of the differences in surface properties between chalcopyrite and the gangue. During the process, collectors such as xanthates turn chalcopyrite particles hydrophobic, allowing them to adhere to air bubbles and float, while the gangue, being hydrophilic, sinks and is discarded. Modifiers like pH regulators and depressants improve the separation and allow obtaining a more suitable concentrate for the following process [15,16].

Despite recent advances, there are still debates about the best performing reagents, the optimal dosage and the effect of metal ions. Therefore, several studies have focused on analyzing interaction mechanisms and the factors that influence the chemical and physical properties of the mineral’s surface [17,18,19,20]. Analytical techniques such as Fourier transform infrared spectroscopy (FTIR), X-ray photoelectron spectroscopy (XPS) and atomic force microscopy (AFM) have provided key experimental information [19,21], while computational tools, like molecular dynamic simulations (MD) and machine learning (ML), have offered insights at the atomic level [18]. Quantum chemistry, along with calculations like density functional theory (DFT) and periodic boundary conditions (PBCs) have also contributed to describe the atomic and molecular interactions in these systems [22].

It is pertinent to note that molecular dynamics is directly associated with molecular design, since it allows modeling and prediction of atomic level interactions between the reagents and mineral surfaces in the flotation prior to experimental testing. Unlike analytical techniques, which validate these processes, molecular dynamics offers a predictive perspective of their behavior. This does not imply that they should be used separately; on the contrary, their integration allows combining theoretical modeling with experimental validation for a deeper understanding of the phenomenon in question [18].

Therefore, the objective of this paper is to carry out a comprehensive review of the existing literature about chalcopyrite surface interaction mechanisms during froth flotation, with emphasis on the integration of analytical techniques, molecular dynamics simulations and advanced technologies for its understanding. The novelty of this review lies in offering a structured perspective on the evolution of knowledge in this field, incorporating molecular design and smart industry approaches to analyze interactions at the atomic level and optimize reagent development. It is expected that this paper will become a theoretical reference for future research aimed at filling knowledge gaps and for mining companies that intend to innovate in their chalcopyrite flotation and copper recovery processes, making them more selective, efficient and environmentally friendly.

## 2. Materials and Methods

For the literature search, the electronic database “Scopus” was used and relevant keywords such as chalcopyrite flotation, copper recovery, surface characteristics, reagent effect, analytical techniques, molecular dynamics, machine learning and similar terms were used. No restrictions were placed on the year of publication, language or country of origin of the studies. The results of the search queries are presented in Table 1.

The papers found were submitted to a selection process, where the title, abstract and keywords were reviewed, discarding the ones that were repeated or did not relate to the investigation objectives. The papers included were organized in section or analytical categories having in consideration the year of publication, objective, methodology and results. The final number of papers included in the revision was 65 and its classification is presented in Table 2.

The first section explores initial studies on the interaction mechanisms in chalcopyrite flotation, analyzing the relevant process factors. The second section compiles studies that use advanced techniques for the understanding of these mechanisms, the evaluation of existing commercial reagents and the synthetization of new, more selective and sustainable reagents. Finally, the third section includes studies that go one step further, integrating advanced computational tools, such as molecular dynamics simulations and machine learning, to deepen the comprehension of these mechanisms and design customized reagents.

## 3. Relevant Sections

### 3.1. General Information and Initial Investigations (1979–2015)

Chalcopyrite flotation has been widely studied due to its relevance as one of the main production processes for copper recovery in the mining industry. Just like in the case of other minerals, its efficiency depends on a series of operational variables that must be carefully controlled to maximize recovery and selectivity. The first studies focused on the separation of chalcopyrite from other sulfide ores present in the deposits like pyrite, sphalerite and chalcocite, among others. For this purpose, rudimentary froth flotation techniques were employed, using collectors such as xanthate and frothers derived from natural oils. Tests showed that chalcopyrite had a good response to these reagents in slightly alkaline media (between 9 and 11), achieving a relatively simple separation from the gangue [23,24].

Over time, more systematic methods began to be developed to understand the physicochemical principles behind chalcopyrite flotation. Chander explored the electrochemical properties of surface coatings that form on chalcopyrite upon interaction with aqueous solutions, with and without reagents. The results demonstrate that the buoyancy of this mineral is related to the formation of surface coatings. A metal-poor coating adjacent to the mineral can promote flotation even without collectors, while the presence of hydrophilic coatings such as oxides and hydroxides eliminates buoyancy, making reagents necessary. Moreover, it was observed that factors like composition, porosity and oxidation conditions determinate the properties of these coatings. In the presence of xanthates, the formation of products such as copper xanthate and dixanthogen improves the hydrophobicity of the mineral and hence its buoyancy [25].

Senior and Thahar analyzed the impact of metal hydroxides formed in the pulp during flotation and their interaction with collectors on the behavior of chalcopyrite. The results showed that the formation of metal hydroxides, like Fe(OH)_3_ (ferric hydroxide), under alkaline pH conditions reduces the buoyancy of chalcopyrite, as these hydrophilic coatings cover its surface and prevent efficient absorption of the collectors. However, the addition of xanthates can partially counteract this effect by improving mineral hydrophobicity, even in the presence of such coatings [26].

Previous studies have revealed that metal ions present in the pulp influence the mineral’s buoyancy. Therefore, Kant and collaborators investigated the metal ions’ distribution on the chalcopyrite surface among the different products generated during the flotation process, including concentrate, froth and tailings. The results showed that copper ions (Cu^2+^) predominate on the surfaces of the concentrate particles, while iron ions (Fe^3+^) are found mostly in the tailings, indicating that hydrophilic particles tend to retain more iron. In addition, the study confirmed that metal ions not only affect the surface chemistry of the mineral but also influence the selectivity of the flotation process [27].

Another key factor in chalcopyrite flotation is the upstream comminution process, encompassing particles grading and milling conditions. Peng and collaborators showed that finer particles tend to float better due to their higher specific surface area, which increases interaction with reagents, although they cautioned that overgrinding could generate ultrafine particles that are difficult to float, reducing efficiency. Moreover, they pointed out that the use of chromium steel milling media minimizes the oxidation of the mineral surface, reducing the formation of hydrophilic products such as oxides and sulfates on the chalcopyrite, improving its buoyancy. In contrast, mild steel milling media generates better oxidation, activating the pyrite and complicating its separation [28].

In this regard, Peng and Grano observed that fine particles are much more prone to oxidize and absorb iron oxidation species, decreasing their buoyancy, while intermediate particles are less susceptible to this effect. They concluded that surface oxidation increases the affinity of fine particles for iron species, which explains their depression on flotation. They recommended the use of more inert milling conditions, such as chromium steel media and reducing atmospheres, to minimize the negative effects of oxidation and improve fine particles recovery from chalcopyrite [29].

Electrochemical parameters, such as oxidation-reduction potential (Eh) and its influence on chalcopyrite flotation, have also been of interest to the literature. Going back a little, He and collaborators noted that, at certain Eh values, hydrophilic coatings form on the pyrite surface, which decreases its buoyancy and improves selectivity towards chalcopyrite [30]. Along the same lines, the effects of aeration and pyrite content were studied. Experiments, which included measurements at different aeration levels, showed that high pyrite levels and excessive aeration elevate the Eh, promoting the formation of metal hydroxides on the chalcopyrite surface, reducing its buoyancy [31,32].

The significant role of reagents in chalcopyrite flotation has been mentioned above. For that reason, several researchers have concentrated their efforts on the development of collectors, depressants, frothers and others that allow achieving an efficient and selective flotation, especially in the presence of gangue and other undesirables. After an exhaustive review of the collectors available at the time, Bulatovic highlighted the efficiency of xanthates and dithiocarbamates in chalcopyrite recovery, indicating that they absorb on the mineral surface, increasing its hydrophobicity and facilitating its separation [15].

The use of selective depressants also became popular at that time. Reagents like NaCN (sodium cyanide) and ZnSO_4_ (zinc sulfate) were used to inhibit the flotation of unwanted minerals, such as pyrite, which allowed clean and efficient chalcopyrite recovery and, consequently, an improvement in the quality of copper concentrates [33,34].

In relation to frothers, we cannot leave aside the work of Leja, who describes the impact of frothers on surface tension and the formation of stable bubbles [35]. In this regard, Fuerstenau and collaborators investigated the influence of different types of frothers, such as alcohols and polyglycols, on bubble stability and bubble size during chalcopyrite flotation. Their findings indicated that the choice of frother directly affects process efficiency, as it determines froth quality and the recovery of the desired mineral [16].

Over time, it became evident that traditional reagents are not always sufficient to achieve efficient and selective chalcopyrite flotation, especially in the presence of pyrite. As will be seen in the following sections, this led to the development of new reagents and surface modifiers, accompanied by innovative proposals for the optimization of operating conditions.

In general, initial research on chalcopyrite flotation allowed the identification of factors influencing mineral buoyancy, such as pH, reagents used, formation of surface coating and milling conditions. These findings, although limited by the tools of the time, were crucial for the development of more efficient practices and established the theoretical and experimental foundations that support the optimization of modern copper recovery processes.

### 3.2. Analytical Techniques and Innovative Reagents (2009–2024)

Despite the studies conducted on the details of chalcopyrite flotation, there is still uncertainty in regard to specific mechanisms that regulate the interaction between the mineral surface and the reagents used to optimize its separation from the gangue. For this reason, numerous investigators have carried out experimental tests, using various analytical techniques and reagents, both commercial and synthesized, to study the physicochemical behavior of the process. Table 3 presents a summary of the most recurrent standard measurement techniques in the literature.

Liu et al. investigated the use of calcium lignosulfonate (LSC) as an organic depressant in the flotation separation of chalcopyrite and pyrite. Using flotation tests and FTIR spectrometry, they demonstrated that LSC effectively reduces pyrite buoyancy over a wide pH range, without significantly affecting the flotation of chalcopyrite. The analysis showed that LSC is absorbed on the pyrite surface, decreasing its hydrophobicity by preventing the formation of dixanthogen. In relation to chalcopyrite, the collector butyl xanthate absorbs preferentially, limiting LSC interaction. It was concluded that LSC allows an efficient selective separation between both minerals under alkaline conditions [36].

Mierczynska and Beattie studied the adsorption of various carboxymethylcellulose (CMC) polymers on talc and chalcopyrite, analyzing their effect on flotation. They used AFM, contact angle measurements and flotation tests to evaluate the interaction between the polymers and the minerals’ surfaces. They discovered that the degree and distribution of carboxymethyl groups in the polymers influence covering ability, which reduces the hydrophobicity of the minerals. Polymers with low substitution and uniform distribution showed better efficacy as depressants, highlighting the CMC LSLB for its performance in reducing talc buoyancy. It was concluded that these differential properties are key for the design of selective reagents in flotation processes [37].

Wang et al. investigated the selective flotation of chalcopyrite relative to pyrite, using sodium glycerylxanthate (SGX) as a depressant. Results showed that SGX significantly reduced pyrite buoyancy over a pH range of 7 to 10, while the chalcopyrite flotation was hardly affected. This was attributed to the strong adsorption of SGX on the pyrite surface, forming a hydrophilic film that inhibited the action of the sodium butylxanthate (SBX) collector. Zeta potential measurements confirmed a higher adsorption of SGX on pyrite compared to chalcopyrite, especially in alkaline conditions. In addition, FTIR analysis revealed that hydroxyl groups of SGX contributed to pyrite hydrophily, which explains its depression on flotation. It was concluded that SGX is an efficient and environmentally friendly depressant for the selective separation of copper sulfide ores [38].

He and collaborators synthesized a new surfactant called N,N–diethyl–N′–cyclohexylthiourea (DECHTU), and evaluated it as a collector in chalcopyrite flotation. Microflotation tests demonstrated that, in DECHTU presence, chalcopyrite exhibited good hydrophobicity and was effectively concentrated with nitrogen bubbles in a pH range between 4 and 8. Adsorption data fitted the kinetic model of pseudo second order and Langmuir isotherm, indicating a spontaneous and exothermic chemisorption process. Zeta potential analyses suggested that DECHTU adsorbs anionically, releasing H+ in solution, while XPS spectroscopy confirmed the formation of surface Cu(I)–DECHTU complexes and the reduction of cupric to cuprous copper. It was concluded that DECHTU enhances chalcopyrite flotation [39].

Qu et al. synthesized another surfactant called 3–hexyl–4–amino–1,2,4–triazole–5–thione (HATT) to study its behavior as a collector and adsorption mechanisms in chalcopyrite flotation. They performed microflotation tests, zeta potential analysis, FTIR spectroscopy and the measurement of adsorption quantities. The microflotation test results and zeta potential analysis showed that HATT presented a reliable performance in a pH range between 4 and 8. Also, the adsorption of the compound on the chalcopyrite surface followed a pseudo-second order kinetic model and was characterized as a spontaneous, endothermic and chemisorption process, according to the thermodynamic analyses. FTIR spectroscopy studies suggested the formation of complexes between the sulfur and nitrogen atoms of HATT and the copper atoms present in chalcopyrite [40].

Peng and collaborators investigated the effects of the addition of sodium butyl xanthate (SBX) before and after milling on chalcopyrite flotation, as well as the surface properties of the mineral. In the study, the addition of SBX before milling showed a better performance in chalcopyrite recovery, reaching 93.62% versus 90.03% obtained by adding it afterwards. This result was attributed to higher pump potential and better adsorption of the collector on fine particles (<20 μm), which increased hydrophobicity. In addition, XPS analyses showed that a higher amount of adsorbed collector, higher free oxygen adsorption and lower presence of oxidized iron species on the mineral surfaces under the pre-addition method. It was concluded that, with sufficient doses of SBX, the addition of collector prior to milling optimizes chalcopyrite recovery [41].

Xiao et al. synthesized the surfactant O–isopropyl–S–[2–(hydroxyimino) propyl] dithiocarbonate ester (IPXPO) and introduced it as a collector for chalcopyrite flotation. Its behavior and adsorption mechanisms were investigated by microflotation tests, zeta potential, thermodynamics and adsorption kinetics. It was found that IPXPO presented superior performance in chalcopyrite flotation compared to pyrite, operating efficiently in a pH range between 4 and 9. The adsorption mechanism followed the Langmuir isotherm and pseudo-second order kinetic model, indicating a spontaneous endothermic chemisorption process. As shown in Figure 1, SECM images and FTIR analyses showed the formation of IPXPO–Cu surface complexes through Cu–S, Cu–N and Cu–O bonds, facilitated by the C=S and –C=N–OH functional groups of the surfactant. It was concluded that IPXPO is a very efficient collector for chalcopyrite separation [42].

Figure 1 shows the formation of Cu–S, Cu–N and Cu–O bonds is observed, and SECM images show the changes before and after treatment, evidencing the formation of new compounds that explain the high selectivity [42].

He et al. synthesized an emulsion collector based on hydrophobic polystyrene nanoparticles with thiazole functional groups (HNP) by emulsion polymerization to improve microfine chalcopyrite recovery in flotation processes. Flotation tests showed that the HNP collector has a high selectivity towards chalcopyrite in acid and neutral media, reaching a recovery higher than 95% at pH 6. Through FTIR spectrometry and zeta potential analysis, it was demonstrated that the adsorption of HNP on the chalcopyrite surface is chemical in nature, favoring the mineral’s buoyancy. In addition, images obtained with SEM microscopy confirmed the strong selective adsorption of HNP on chalcopyrite, with a uniform distribution and an average particle size of 77 nm. It was concluded that the HNP collector, being anionic, is effective for selective chalcopyrite flotation, especially for microfine mineral separation [43].

Liu and collaborators investigated the interaction mechanism of a mixture of ZnSO_4_ (zinc sulfate) and sodium dimethyldithiocarbamate (SDD) in the differential flotation of Cu–Zn sulfide minerals. Through microflotation experiments, they determined that this mixture has a significant selective depressant effect on sphalerite, without affecting chalcopyrite flotation. With a ZnSO_4_:SDD dosage in a 3:1 ratio, a butyxanthate (BX) collector concentration of 10^–5^ mol/L and a pH adjusted to 10, a 30.21% Cu concentrate was obtained with an 86.79% recovery. Moreover, a Zn content of 4.20% with a 5.48% recovery was obtained. Zeta potential analysis and LEIS spectroscopy results confirmed that the mixture restricts the adsorption of BX on the sphalerite surface compared to chalcopyrite, especially in an alkaline pH range (9–12), which explains the excellent buoyancy of the latter under these conditions [44].

Chen et al. investigated the use of sodium alginate as a depressant in the selective chalcopyrite flotation versus galena, using dibutyl ammonium dithiophosphate as a collector. Through microflotation, zeta potential, FTIR spectroscopy and XPS analysis tests, they demonstrated that sodium alginate adsorbs selectively on the galena surface, decreasing its buoyancy while maintaining chalcopyrite buoyancy. This behavior is attributed to chemical interactions between Pb^2^⁺ ions and the alginate functional groups, which block the adsorption of the collector. On the other hand, alginate does not significantly affect the surface of chalcopyrite. Based on these results, it was concluded that the sodium alginate is an effective, non-toxic and environmentally friendly depressant for the separation of Cu–Pb minerals in flotation systems [45].

Chen et al. evaluated the use of a seaweed-derived adhesive, called seaweed glue (SEG), as a novel polymeric depressant, along with butylxanthate (BX) as a collector, for the selective separation of chalcopyrite and galena by flotation. Through microflotation experiments, contact angle analysis, adsorption, dynamic potential and FTIR spectroscopy, they concluded that SEG has a stronger depressant effect on galena than on chalcopyrite. This is achieved at pH 8.0, with an SEG concentration of 15 mg/L, obtaining a Cu concentrate with a 23.68% grade and a recovery of 81.52%. The results indicated that SEG interacts with galena through strong chemisorption, while its interaction with chalcopyrite is based on weak physical absorption [46].

Huang et al. investigated the synthesis and application of S–hydroxyethyl–O–isobutyl xanthate (HEIBX) as a selective collector in chalcopyrite and pyrite flotation. Using microflotation tests and techniques such as density functional theory (DFT), contact angle measurement and zeta potential, they demonstrated that HEIBX improved chalcopyrite recovery to 93.94% and reduced pyrite buoyancy to 11.36% compared to sodium isobutyl xanthate (SIBX). FTIR, XPS and UV–Vis analyses indicated that HEIBX is adsorbed chemically on chalcopyrite by forming of Cu–S and Cu–O bonds, while adsorption on pyrite is weak. The incorporation of a hydroxyethyl group on HEIBX increased its reactive site density and electron donating capacity, improving its performance. As shown in Figure 2, HEIBX is a more efficient and selective collector for chalcopyrite separation at neutral pH conditions [47].

Jia and collaborators investigated the flotation ability of a surfactant called thiohexanamide (THA) as a selective collector for chalcopyrite separation from pyrite and galena. The study included microflotation and bench scale tests, zeta potential analysis and DFT calculations to understand the adsorption mechanisms. The results demonstrated that THA exhibits a stronger affinity towards chalcopyrite than traditional collectors such as SIBX and IPETC, achieving a recovery of 97.1% for chalcopyrite at pH 8. FTIR and XPS analyses confirmed that THA interacts chemically with chalcopyrite forming Cu–S and Cu–N bonds, while it showed weak interactions with pyrite and galena. DFT calculations revealed that the functional groups of THA, particularly C=S and NH2, favor its high selectivity and reactivity towards chalcopyrite, positioning it as a promising collector for chalcopyrite separation in complex mixtures [48].

Pan et al. studied the effects of sodium alginate as a selective depressant in chlorite and serpentine flotation, common minerals in copper sulfide deposits. Microflotation tests showed that sodium alginate reduces the buoyancy of these magnesium silicates, while keeping the recovery of chalcopyrite high. The results of the ternary mixtures flotation experiments showed that, at a pH9 and a dose of 40 mg/L of sodium alginate, a concentrate with 31% of copper and a recovery of 90% was obtained. Through adsorption tests, zeta potential and FTIR spectroscopy, it was concluded that the depression mechanisms of the sodium alginate involve chemical adsorption on the chlorite and serpentine surfaces, increasing their hydrophilicity and generating electrostatic repulsion. This effect was not observed in chalcopyrite, which confirms its selectivity as a depressant [49].

Yuan and collaborators investigated the use of O–Carboxymethyl chitosan (O–CMC), a non-toxic and biodegradable chitosan derivative, as a selective depressant for molybdenite in the separation of copper and molybdenum sulfides. Flotation tests showed that O–CMC can effectively depress the molybdenite flotation in a pH range of 3 to 11, while chalcopyrite recovery was not significantly affected, except under acidic conditions at pH 3. Through AFM microscopy imaging tests, a strong adsorption of O–CMC to the molybdenite surface was observed, in contrast to chalcopyrite, where no significant adsorption was detected after washing. These results indicate that differential adsorption is responsible for the selective separation achieved [50].

Chimonyo et al. synthesized an oxidized starch, named Ox 5/120, and used techniques such as AAS spectroscopy and AFM microscopy to investigate its impact on the differential flotation of chalcopyrite and graphite. They found that oxidized starch showed significantly higher depressive capacity on graphite, with modest effects on chalcopyrite, than native starch. This behavior was attributed to the generation of carboxyl and carbonyl functional groups, as well as depolymerization that increased solubility and modified the molecular conformation. In addition, the images obtained revealed a more flexible organization of the polymeric chains on the graphite surface, increasing its hydrophobicity [51].

Gutiérrez and collaborators investigated the use of lignosulfonates to separate chalcopyrite and molybdenite by flotation, with the objective of developing more environmentally friendly processes than the use of NaHS. Low molecular weight lignosulfonates were synthesized by kraft lignin sulfomethylation and compared with commercial lignosulfonates. Chemical characterization included elemental analysis, FTIR spectroscopy, gel permeation chromatography and molecular masses measurements. Results showed that the lignosulfonates significantly depress molybdenite flotation, an effect that increases with pH due to specific interactions with calcium metal sites on the mineral surface. In chalcopyrite’s case, a strong depression was observed using certain collectors, although this effect was mitigated by using long-chain collectors such as potassium amyl xanthate [52].

Jia et al. synthesized three new thioxopropanamide-type surfactants: (a) 3-ethylamino-N-methyl-3-thioxopropanamide (EAMTXPA), (b) N,N-dimethyl-3-(phenylamino)-3-thioxopropanamide (PhATXPA) and (c) 3-ethylamino-N-phenyl-3-thioxopropanamide (EAPhTXPA), which were evaluated as collectors for chalcopyrite flotation. Through DFT calculations, the chemical reactivity sites of the surfactants were determined, highlighting that EAPhTXPA has higher electron donating capacity and better hydrophobicity. Microflotation tests showed that with EAPhTXPA a maximum recovery of 97.5% was achieved at an optimum pH of 8. Also, contact angle analysis showed that the hydrophobicity of chalcopyrite increased significantly after treatment with EAPhTXPA. Adsorption tests and XPS spectroscopy confirmed that the adsorption process is a spontaneous and endothermic chemisorption, mediated by the formation of Cu-S, Cu-O and Cu-N bonds on the mineral surface. As seen in Figure 3, EAPhTXPA has superior performance as a chalcopyrite collector. Bonds are observed in (I) Cu–S, (II) Cu–O and (III) Cu–N, indicating chemisorption that improves the hydrophobicity of the mineral [53].

Liu and collaborators synthetized 6–Hexyl–1,2,4,5–tetrazinane–3–thione (HTT), a novel collector for chalcopyrite and pyrite flotation. Through microflotation tests, HTT was shown to have a chalcopyrite recovery of 93.5% at pH 10.5, while the pyrite flotation was less than 35%, outperforming the traditional collector sodium hexyl xanthate (SHX). AFM images and contact angles showed higher hydrophobicity on the surface of chalcopyrite treated with HTT. UV spectra and DFT calculations indicated that HTT has a high affinity for cupric and cuprous ions, forming Cu-N and Cu-S bonds on the chalcopyrite surface, while its interaction with iron ions was limited. XPS results confirmed the chemisorption of HTT on chalcopyrite, highlighting its superior selectivity and efficiency for the separation of copper sulfide minerals [54].

Duan and collaborators synthesized the N-benzoyl-N′,N′-diethyl thiourea (BDETU) collector by the reaction of benzoyl chloride, potassium thiocyanate (KSCN) and diethylamine, confirming its structure by NMR and FTIR. In flotation experiments, BDETU achieved a chalcopyrite recovery of 96.50% at pH 8.0, outperforming IPETC, while pyrite recovery was 20.65%. Adsorption studies indicated that BDETU interacts strongly with chalcopyrite, increasing its hydrophobicity, while in pyrite the interaction was weak and without significant effect. UV–Vis analysis, FTIR and DFT calculations showed that BDETU is chemically adsorbed on chalcopyrite by C=O and C=S groups, forming C-O-Cu and C-S-Cu bonds with high affinity towards Cu+ and Cu2+, with no relevant interaction with Fe^2+^ and Fe^3+^. It is concluded that BDETU is a selective and efficient collector for chalcopyrite flotation [55].

He and collaborators synthesized a new diminerophilic group collector, 5–methyl isobutylxanthate–1,3,4–oxadiazole–2–thione (MIXODT), for chalcopyrite flotation. Through microflotation experiments, contact angle measurements and zeta potential, the hydrophobic properties of the mineral treated with this collector were evaluated. The results indicated that MIXODT had higher flotation capacity than the conventional collector sodium isobutyl xanthate (SIBX) in a pH range from 3 to 9, reaching over 95% recovery under optimal conditions. The FTIR and XPS analyses showed that MIXODT adsorbs chemically on the chalcopyrite surface through the formation of Cu–S and Cu–N bonds, accompanied with the reduction of Cu(II) to Cu(I). In addition, density functional theory calculations confirmed that the functional groups of MIXODT, such as thiocarbonyl and oxadiazole thione groups, are responsible for interaction with copper ions [56].

Zhang et al. used individual mineral flotation experiments and collector adsorption measurements to investigate the effect of sodium polyaspartate (PASP) as a galena depressant on chalcopyrite flotation. They found that the addition of PASP at a concentration of 10 mg/L, under weak alkaline conditions of pH 10, maintained a chalcopyrite recovery of 91.9%, while galena recovery decreased to 1.1%. FTIR and XPS analysis showed that PASP absorbs chemically on the surface of both minerals, interacting more strongly with galena. The results suggest that competitive adsorption between PASP and collectors, such as ammonium dibutyl dithiophosphate (ADD), decreases galena buoyancy [57].

Zou et al. synthesized a novel surfactant named O,O′-bis(2-butoxyethyl) ammonium dithiophosphate (BEAT), and combined it with O-isopropyl-N-ethyl thionocarbamate (IPETC) to improve the recovery of chalcopyrite by flotation. The study used microscale and bench scale flotation tests, surface property analysis, wettability measurement and molecular interaction parameter calculations to evaluate the performance of this mixed reagent scheme. Results showed that a 60% mole fraction of BEAT in the mixture achieved a maximum chalcopyrite recovery of 92.43% over a pH range of 7.0 to 7.5. FTIR analyses confirmed the chemical adsorption of the collectors on the chalcopyrite surface. In conclusion, the mixed IPETC/BEAT collectors demonstrated higher surface activity, selectivity and hydrophobicity compared to the individual collectors, representing a promising scheme to improve the flotation efficiency of chalcopyrite against gangue minerals such as pyrite, with the additional benefit of reducing costs due to less use of reagents [58].

He and collaborators conducted a study on selective depression mechanisms of galena using tea polyphenol (TP) and sodium isoamyl xanthate (SIX) in chalcopyrite flotation. The mixed minerals flotation tests, performed at pH of 8 and concentrations of 4 × 10^–4^ mol/L TP and 2 × 10^–4^ mol/L SIX, reached a copper recovery of 83.07% with a grade of 29.32% in the copper concentrate, while the lead recovery was of 8.87% with a grade of 7.87%. Zeta potential analysis and adsorption tests indicated that TP significantly inhibits the adsorption of the SIX collector on the galena surface, without greatly affecting the adsorption on the chalcopyrite surface. ToF–SIMS spectrometer and XPS analysis results confirmed that TP chemically adheres to the Pb elements in galena, increasing its hydrophobicity and making difficult the collector adsorption [59].

Jiang et al. investigated the use of dibutyl phosphonate (HDBP) as a selective collector for the flotation of chalcopyrite versus galena and pyrite. Microflotation tests showed that HDBP allows a recovery higher than 85% of chalcopyrite at low concentrations and in a pH range between 6 and 8, while galena and pyrite recoveries remained below 20%. XPS analysis indicated the formation of P-O-Cu and P=O-Cu chemisorption bonds between HDBP and chalcopyrite, supported by DFT simulations that showed high adsorption energies for chalcopyrite compared to galena and pyrite. Electrochemical tests confirmed a strong interaction of HDBP with chalcopyrite, evidenced by significant increases in corrosion potential and current density. As shown in Figure 4, HDBP is a highly selective collector and potentially applicable for the separation of copper sulfide minerals in complex systems [60].

Wang and collaborators introduced ethylenediamine tetramethylenephosphonic acid (EDTMPA) as an innovative collector to improve the selective separation of chalcopyrite from pyrite under low alkalinity conditions. Flotation test results, using binary mineral mixtures and real ore, demonstrated that EDTMPA outperforms conventional collectors such as ethyl xanthate (EX) in terms of selectivity, reaching 88.36% chalcopyrite recovery and 13,93% pyrite recovery over a wide pH range of 6 to 11. Moreover, electrochemical and FTIR spectroscopy studies confirmed that EDTMPA absorbs more strongly and specifically on chalcopyrite than on pyrite. Density functional theory (DFT) calculations revealed that the phosphonic groups of EDTMPA form more stable chemical bonds with Fe and Cu atoms on the chalcopyrite surface, highlighting its selective affinity. It was concluded that EDTMPA, due to its lower toxicity, lack of odor and high efficiency, represents a potential for industrial applications in chalcopyrite flotation, reducing the dependency of highly alkaline processes and improving environmental sustainability [61].

Perera et al. synthesized a new cardanol derivative named 3-pentadecylphenyl 4-(3,3-diethylthiouredo-4-oxobutanoate) (DP089), to evaluate its flotation efficiency and surface adsorption mechanism on chalcopyrite and pyrite. This compound, designed with functional groups such as C=O and -NH-C(=S)-N-, showed remarkable selectivity towards chalcopyrite compared to the conventional collector potassium amyl xanthate (PAX). Flotation experiments revealed that DP089 reached a maximum chalcopyrite recovery of 92.1% at pH 8, while pyrite recovery was significantly lower. UV–Vis and FTIR analyses showed that DP089 forms chemical bonds as C-O-Cu and C-S-Cu with Cu+ ions present on the chalcopyrite surface, which increases its hydrophobicity. In contrast, no significant interactions were observed with Fe2+ ions in pyrite. It was concluded that the high efficiency of DP089 is due to its ability to selectively chemosorb on chalcopyrite, optimizing the flotation separation process [62].

He and collaborators synthesized the 5–methyl diethyl dithiocarbamate–1,3,4-oxadiazole–2–thione (MDCODT) collector and introduced it in the selective flotation of chalcopyrite and molybdenite. Microflotation results showed that this collector allows an effective separation of the minerals without the use of inhibitors, reaching a chalcopyrite recovery up to 90% at pH 9. Zeta potential tests, UV–Vis spectrometry and adsorption analysis demonstrated that MDCODT absorbs preferentially on chalcopyrite, forming chemoadsorbed coatings that increase its hydrophobicity. In addition, FTIR and XPS spectrometry studies, along with DFT calculations, confirmed the formation of Cu–S and Cu–N bonds on the chalcopyrite surface [63], as shown in Figure 5.

Liang and collaborators investigated the mechanisms through which calcium lignosulfonate selectively depresses fine molybdenite flotation during Cu–Mo flotation separation, using ethyl potassium xanthate as a collector. Through flotation tests and FTIR and XPS spectroscopy, it was determined that calcium lignosulfonate significantly reduces the hydrophobicity of molybdenite by physically interacting with its surface, while in chalcopyrite the interaction is chemical, via chelation and complexation between the oxygenated groups of lignosulfonate and the exposed metal sites on its surface. It was observed that ethyl potassium xanthate has a desorbing effect on the lignosulfonate adsorbed on chalcopyrite, restoring is buoyancy, but with a much smaller effect on molybdenite, thus allowing selective separation. It was concluded that calcium lignosulfonate is an efficient and sustainable depressant for molybdenite in Cu–Mo separation [64].

Perera et al. developed a new collector via RAFT polymerization, named poly(CA_4_–co–ACOEA_14_), functionalized with O-ethyl ace-tylcarbamothioate and cardanol, combining specific hydrophobic and functional characteristics for flotation. They studied the selective separation of chalcopyrite using adsorption tests, UV, FTIR and XPS spectroscopy, and contact angle measurements to evaluate the ability of the polymer compared to a xanthate-based one. The results demonstrated a higher affinity of the designed polymer towards chalcopyrite over pyrite, attributed to its selective chemisorption ability, which also increases its hydrophobicity. In addition, the polymer showed dual behavior, functioning as a collector and flocculant, improving the aggregation of fine chalcopyrite particles. The efficiency of the new reagent in selective flotation and tailings management was highlighted, promoting more sustainable mining practices [65].

Sun et al. investigated the selective flotation of chalcopyrite versus pyrite and sphalerite using a novel collector, O–isobutyl–N–allyl tionicarbamato (IBATC). Through flotation of individual minerals, adsorption analysis, zeta potential, XPS and FTIR spectroscopies and DFT calculations, they demonstrated that IBATC has a higher affinity for chalcopyrite in the pH range of 9.5–10, reaching a recovery of more than 80% with concentrations between 30 to 40 mg/L. As is observed in Figure 6, IBATC absorbs chemically on chalcopyrite through interactions between the S atoms of the collector and surface Cu, while on pyrite and sphalerite case the absorption is physical. DFT results confirmed the spontaneous and exothermic electron transfer between the S of IBATC and the Cu of chalcopyrite, which improves its selectivity and efficiency against conventional collectors [66].

Sun and collaborators synthesized O-methylisobutyl-N-allyl thionocarbamate (MIBATC), a novel collector for the separation of chalcopyrite from sphalerite and pyritite. Flotation experiments showed that MIBATC achieved more than 80% recovery of chalcopyrite over a pH range of 9.5 to 10 and a concentration of 30 to 40 mg/L, with low recoveries of sphalerite and pyrite. Zeta potential and adsorption amount analyses indicated a strong affinity of MIBATC for chalcopyrite, while its interaction with sphalerite and pyrite was limited. FTIR, XPS and MNR studies revealed a chemisorption of S from MIBATC to the Cu of chalcopyrite, forming Cu-S bonds and eliminating oxidation products. DFT calculations confirmed that the interaction was a spontaneous and exothermic reaction, with the S of MIBATC being the main reactive site. The efficiency and selectivity of MIBATC is highlighted, promoting its application [67].

Wang and collaborators investigated the selective flotation mechanism of chalcopyrite versus pyrite by using a novel chelating collector with diminerophilic group, named 5–Methyl Butylxanthate–1,2,4–Triazole–3–Thione (MBuXTT). Through microflotation experiments, contact angle analysis, zeta potential, UV–Vis, FTIR and XPS spectroscopy and DFT calculations, the authors demonstrated that MBuXTT improves the separation quality of chalcopyrite. The collector showed a higher affinity toward copper on the chalcopyrite surface, forming Cu(I)–S and Cu(I)–N bonds via a spontaneous and endothermic chemisorption process. The results also revelated that MBuXTT outperforms traditional collectors such as sodium butyl xanthate in terms of selectivity and efficiency, while minimizing environmental impact by reducing the need for inhibitors [68].

As a synthesis, Table 4 presents a comparison of the reagents used, the analytical and experimental methods applied and the results obtained in each of the studies included in this section.

### 3.3. Molecular Design and Machine Learning (2018–2024)

Molecular dynamic (MD) is consolidating as a key computational tool in the analysis of complex molecular interactions, being useful in the understanding of several industrial and mining processes, such as the case of chalcopyrite flotation. It allows, among other things, modeling and simulating the behavior of molecules and mineral surfaces at the atomic level, providing detailed information about the mechanisms ruling the adsorption and interaction between reagents and minerals.

The above, along with the application of specialized software, has allowed investigators to evaluate various flotation conditions, from pH changes to variations in the reagent’s concentration. Theorical simulations and their validation though experimental tests have made it possible to develop suggestions for improving flotation processes with a more precise and customized approach. Table 5 presents some software frequently used in literature.

Bu et al. laid the foundations for using MD to study chalcopyrite flotation by analyzing how the collector O-Isopropyl-N-Ethyl Thionocarbamate (IPETC) interacts with chalcopyrite and galena surfaces. Simulations showed that IPETC adsorbs stably on both surfaces, although it has a higher affinity for galena, supporting its ability to overcome water interactions and improve buoyancy. In addition, the study showed how pH increase affects these interactions due to competition with hydroxyl OH- ions, setting a precedent in the evaluation of buoyancy conditions by MD [69].

In a later development, Dai et al. used MD to model the interactions between the peracetic acid (PAA) depressant and minerals such as arsenopyrite and chalcopyrite. Simulations revealed that PAA absorbs more strongly and selectively on arsenopyrite, due to its oxygen atoms forming covalent bonds with Fe atoms, with short bond distances, overcoming the interaction with chalcopyrite. In addition, it was observed that PAA induces in the Fe(II) to Fe(III) oxidation in arsenopyrite, forming Fe(III)–O species that increase hydrophobicity and reduce the affinity of collectors like xanthate. These findings support the use of PAA as an efficient selective depressant to separate arsenopyrite from chalcopyrite and provide valuable information for the development of more effective and sustainable depressant agents [70].

Yang et al. expanded the MD field by exploring the interaction between poly styrenebutyl acrylate (St-Ba) nanospheres and chalcopyrite. The results showed that the interaction energy was negative, suggesting a spontaneous and exothermic adsorption process. During the simulation, the molecular chains of the nanospheres were uniformly distributed on the mineral surface, generating a strong electrostatic and hydrophobic interaction that increased the hydrophobicity of chalcopyrite. This behavior explains the bridging-type mechanism of the nanospheres, which facilitated the adherence of chalcopyrite particles to air bubbles, increasing their buoyancy even in the presence of serpentine, a mineral that typically inhibits this process. The findings revealed new possibilities for the design of non-traditional collectors with higher efficiency [71]. Figure 7 presents an MD simulation of the interaction between the St-Ba molecule with the chalcopyrite surface.

Dong et al. furthered on MD and DFT calculations to study the adsorption mechanism of depressants such as sodium thioglycallate (STG) on arsenopyrite and chalcopyrite surfaces in the presence of water, simulating real condition in an aqueous media. The findings demonstrated that the –SH group of STG forms chemical bonds with the Fe and As active sites on arsenopyrite, while the –COO– group forms hydrogen bonds with water molecules, generating a stable hydrophilic film on this mineral surface, drastically reducing its buoyancy. This behavior was not observed in chalcopyrite, highlighting the selectivity of STG on the mineral separation on moderately alkaline conditions [72].

Lin and collaborators used MD to study the interactions of the collector O-isobutyl-N-hydroxyethyl thionocarbamate (IBHETC) with metal ions and mineral surfaces such as chalcopyrite. The results showed that IBHETC stably adsorbs by forming N-Cu, O-Cu and S-Cu bonds, stabilized by four and five-membered rings. During this interaction, IBHETC undergoes deprotonation and isomerization, with the N and O atoms acting as electron donors and the S-Cu bond exhibiting positive coordination characteristics. The inclusion of the hydroxyl group significantly improves the adsorption and selectivity of the collector towards chalcopyrite versus pyrite, highlighting the effectiveness of IBHETC for selective flotation and the ability of MD to predict its stability under specific conditions [79].

Liu and collaborators also advanced MD by comparing the hydrophobicity induced by dibutyl dithiophosphate (DTP) and 3418A collectors on chalcopyrite. Simulation results showed that 3418A has a more negative adsorption energy than DTP, indicating a stronger interaction with chalcopyrite. As shown in Figure 8, the hydrophobicity of chalcopyrite surface after the adsorption of 3418A is higher than that of DTP, which improves its buoyancy. It was also found that the Cu–S bonds formed by 3418A are shorter and have a higher Mulliken charge, which suggests a higher covalent stability. It is concluded that 3418A performs better than DTP in chalcopyrite flotation processes, highlighting the usefulness of MD for selecting collectors based on specific molecular characteristics [81].

Gao et al. demonstrated through MD how oxalic acid (OA) and carboxymethyl cellulose (CMC) operate as talc depressants on chalcopyrite flotation. Simulations revealed that OA improves the CMC adsorption on the talc surface, forming dense hydrophilic coatings that reduce its buoyancy and improve selectivity. SEM and FTIR analyses confirmed this higher retention, validating the efficiency of the OA–CMC system [73].

Simultaneously, Shen and collaborators investigated via MD the effect of pyrogallic acid (PA) as a depressant in chalcopyrite flotation. Simulations showed that PA is strongly absorbed in galena, increasing its hydrophobicity and reducing its buoyancy, while chalcopyrite is less affected. IR spectroscopy identified the functional groups responsible for the adsorption, and SIMS and XPS analyses confirmed an increase on the Pb–O and (SO_4_)^2–^ hydrophilic species in galena. The results validate PA as an efficient ecological depressant under light alkaline conditions [74].

In a similar work, Shen et al. analyzed the depressant mechanism of mercaptosuccinic acid (MSA) in the flotation separation of galena and chalcopyrite. MD simulations showed that MSA adsorbs on the surface of galena, forming a stable hydrophilic film by increasing water adsorption, while its effect on chalcopyrite is limited. The -COOH and -SH functional groups of MSA have high affinity for Pb active sites, reducing the buoyancy of galena. FTIR spectroscopy confirmed this differential adsorption, XPS analysis evidenced the increase of Pb-O components and contact angle and AFM measurements demonstrated higher wettability and changes in the surface morphology of the galena. All these results validate MSA as an effective and selective depressant in alkaline conditions [18].

Additionally, Shen and collaborators explored the use of polymaleic acid (PMA) as a galena depressant in chalcopyrite flotation. MD simulations revealed that PMA absorbs more strongly on galena, increasing the accumulation of water molecules on its surface and reducing its buoyancy. FTIR spectroscopy confirmed PMA adsorption, while XPS and SIMS revealed changes in the composition and chemical states of galena, with increased Pb–O and hydroxyl groups. The results validate PMA ability as an environmentally friendly and selective depressor in flotation under alkaline conditions [75].

Yuan and collaborators studied carboxymethyl-β-cyclodextrin (CMCD) as an environmentally friendly depressant for the separation of chalcopyrite and pyrite. MD simulations showed that CMCD has higher affinity for pyrite than for chalcopyrite, forming a hydrophilic layer that blocks adsorption of the SBX collector on pyrite. FTIR and XPS analyses confirmed the interaction of the COO- and -OH groups of CMCD with pyrite, and AFM evidenced higher cohesion with pyrite. The results indicated that CMCD reduces the pyrite buoyancy, highlighting its efficiency as a depressant in chalcopyrite flotation [76].

Zeng et al. applied MD to model the interaction of tannic acid (TA) with chalcopyrite and molybdenite. Simulations showed that TA absorbs more strongly on the molybdenite surface by hydrophobic interactions, while on chalcopyrite it occurs due to chemisorption on Fe sites. FTIR and XPS analyses confirmed these differences, showing Fe–AT bonds in chalcopyrite and the absence of significant chemical interactions in molybdenite. Flotation tests demonstrated that TA blocks adsorption of the PBX collector on molybdenite, reducing its recovery, while allowing adsorption on chalcopyrite. These findings explain TA selectivity as a depressant for Cu–Mo separation [77].

Finally, Zhang et al. employed MD and ab initio molecular dynamics (AIMD) simulations to study the interactions between sodium thioglycolate (STG) depressant and chalcopyrite and molybdenite surfaces. Simulations showed that STG forms Cu(I)-S and Fe(III)-S chemical bonds on chalcopyrite with high adsorption energy that favors its selectivity and increases surface hydrophobicity. On molybdenite, STG showed minimal adsorption, evidencing its specificity. FTIR and XPS techniques confirmed the adsorbed functional groups, while SIMS and UV–Vis spectroscopy measured the adsorption and desorption density, revealing that STG removes up to 81% of the xanthate collector previously adsorbed on chalcopyrite. These findings validate STG as a depressant in the flotation of Cu-Mo minerals [78].

As a synthesis, Table 6 presents a comparison of the reagents used, the methods applied, including the molecular dynamics simulation conditions, and the results obtained in each of the studies included in this section.

In general terms, molecular dynamics has evolved from being a basic tool to study molecular interactions to a key resource in the design of specific and sustainable reagents for mineral flotation. It allows understanding, on a molecular level, the adsorption mechanisms of collectors, depressants and other reagents on mineral surfaces, providing accurate data on affinity, adsorption energy, chemical bond formation and hydrophobic interactions, among others.

The MDs are complemented with analytic techniques like FTIR, XPS, AFM and others presented in the previous section to validate the findings and provide additional details on chemical interactions, surface morphology and hydrophobicity changes. Through their research, the authors have cemented the role of this tool as an indispensable component in the investigation and optimization on chalcopyrite flotation, providing a detailed and predictive framework for future innovations in the mining industry.

In the context of smart industry, emerging technologies are transforming production processes, including those specific to the mining industry, with the aim of making them more efficient. Emphasizing machine learning (ML), its application on chalcopyrite flotation has evolved over the past decade, approaching complex problems related to the characterization, optimization, prediction and sustainability of this process.

At first, investigations such as that of Cook et al. highlighted the ability of ML to address the limitations of conventional methods in chalcopyrite flotation. They proposed a hybrid model based on Random Forest and the FireFly optimization algorithm, called RF-FFA, to predict process efficiency and optimize operating parameters such as reagent dosing, pH, impeller speed and flotation time in real time, maximizing mineral recovery and concentrate quality. The simulations demonstrated that the proposed hybrid model outperforms other individual models, such as artificial neural network (AAN) and support vector machine (SVM), in terms of accuracy and prediction capability in complex and nonlinear systems [82].

Subsequently, He et al. expanded the application of ML by integrating quantum chemistry (QC) into the design of chalcopyrite flotation reagents, reducing cost and time associated with its development. Using QC calculations to obtain accurate data on interactions between a group of candidate molecules and metallic sites, such as Cu(I), Cu(II) and Fe(II), generating key molecular descriptors. Later they applied the Gradient Boosting Regression (GBR) algorithm to predict the selectivity of these molecules, identifying functional groups with high affinity for chalcopyrite and low for other minerals. These findings laid the foundations to guide the design of new, more efficient, selective and environmentally friendly reagents [80].

In recent years, Koh and collaborators have explored the use of deep learning (DL) coupled with advanced instance segmentation techniques to identify and classify mineral particles in high-resolution microscopic images extracted from flotation pulp samples. Their model, which occupies the SOLO v2 architecture in PyTorch, 2.6 allowed real-time determination of particle size and mineralogy without the need for sample preparation, unlike traditional methods such as Mineral Liberation Analyzer (MLA) or Quantitative Evaluation of Minerals by Scanning Electron Microscopy (QEMSCAN). As shown in Figure 9, the results showed that the model can characterize chalcopyrite flotation with high accuracy in minutes, providing a fast and accessible tool for process control. They complemented their findings with XRD and ICP-MS analysis to validate the mineral and elemental composition of the samples [83].

Finally, Zhang and collaborators integrated ML with QC to predict the performance of a group of collectors in the flotation of sulfide minerals, as shown in Figure 10. They characterize through QC calculations a total of 116 collectors and four minerals, including chalcopyrite, galena, pyrite and sphalerite, obtaining descriptors such as surface charge, adsorption energy and electrostatic potential. These data, along with flotation parameters like pH, time and collector concentration, were used as input to train a model based on Extreme Gradient Boosting (XGBoost). These findings demonstrated that this approach allows evaluation of new collectors efficiently, accelerating the design of reagents for the selective flotation of chalcopyrite and other sulfides [22].

As a synthesis, Table 7 presents a comparison of the reagents used, the methods applied, including the characteristics of the machine learning models, and the results obtained in each of the studies included in this section.

## 4. Discussion

This paper presents a comprehensive review of 65 papers on chalcopyrite flotation, from its fundamentals to the application of advanced tools such as molecular dynamics and machine learning. The findings have evidenced that, although the physicochemical principles of flotation are well documented, there are still difficulties in improving its efficiency and selectivity [8,10]. The authors agree that, to address these challenges, the first step is to understand how reagents, chalcopyrite and gangue interact during flotation. Knowing the interaction mechanisms present in the process will allow the development of more precise and sustainable strategies to optimize copper recovery.

The literature also notes that investigation has advanced towards the design of more selective and environmentally sustainable reagents. Studies have demonstrated that the development of new specific collectors, such as HATT, IPXPO and HDBP, improves the hydrophobicity of chalcopyrite without compromising process efficiency [40,42,60]. Similarly, depressants based on natural polymers, like sodium alginate and lignosulfonates, have shown high potential to improve the separation of undesired minerals without affecting copper recovery [36,45,52].

The use of advanced analytical techniques has also been key in the characterization of flotation mechanisms. Techniques such as FTIR spectroscopy, XPS spectroscopy and AFM microscopy have made it possible to analyze the adsorption of the reagents on the mineral surface with accuracy [45,47,50]. These studies have been complemented with computational calculations, highlighting the density functional theory (DFT) and molecular dynamics as novel methods for predicting interactions at atomic level [71,81].

The integration of molecular dynamics and machine learning has enabled significant advances in flotation process modeling. Computational simulation has been shown to be a useful tool for evaluating the adsorption of reagents under different experimental conditions, reducing the need for extensive laboratory testing [19,78]. In particular, machine learning has shown applications in the prediction of reagent performance and in the optimization of flotation operating conditions, suggesting a future oriented towards the automatization of this process [80,82].

An important observation is the wide diversity of experimental conditions used in the analyzed studies. Variations in particle size, pH range, dosage of collectors and depressants, as well as the type of flotation cell employed (e.g., microflotation, Hallimond, Denver), significantly influence the results obtained. For instance, collectors such as IPXPO and HTT showed high chalcopyrite recovery in slightly acidic and neutral media, whereas others like HDBP and THA were more effective under alkaline conditions. These differences reveal that the effectiveness of a reagent is closely linked to the specific operational conditions. Therefore, it is important to standardize certain experimental parameters or, at the very least, report them in detail to allow for a critical and reproducible evaluation of the results.

Regarding molecular dynamics (MD) simulations and density functional theory (DFT) calculations, a disparity in the parameters considered can also be observed. The studies differ in the type of mineral surface modeled, the crystallographic orientation, the number of atomic layers, the type of solvent simulated and the choice of collectors. These decisions affect the accuracy and validity of the predictions. For instance, simulations performed with simplified models may fail to properly capture the interactions of complex collectors, while vacuum simulations or those using implicit solvents may overestimate adsorption energies [53,68]. Likewise, differences in simulation time and periodic cell size impact the stability of the observed configurations. Therefore, it is suggested to adopt more robust protocols that better represent real flotation environments to improve the predictability and applicability of these models.

In terms of economic feasibility, the cost–benefit ratio of the flotation reagents used is a decisive criterion for their industrial implementation. Some innovative collectors, such as those derived from functionalized polymers (e.g., poly(CA4–co–ACOEA14)) or surfactants designed through organic synthesis (such as DP089 and MIXODT), have demonstrated high efficiency and selectivity but may involve high production costs at an industrial scale. In contrast, low-cost reagents like lignosulfonates, sodium alginate or oxidized starch have shown good performance as depressants, particularly due to their low environmental impact and commercial availability.

A key aspect identified in the review is the necessity to continue exploring the interaction between reagents and mineral surface in conditions more representative of the industry environment. Although laboratory studies have enabled significant advancements, industrial scale extrapolation still has its difficulties, especially in terms of reagent stability and the variability of processed mineral characteristics [53,63]. In that sense, the combination of experimental tests with large-scale simulations can contribute to reducing the existing gap between theoretical investigation and practical implementation.

Finally, the environmental impact of the reagents used is still a priority concern. Despite the development of more environmentally friendly alternatives, many commercial reagents still have adverse environmental effects. Future investigations should focus on the progressive replacement of conventional reagents with biodegradable options and the implementation of technologies that minimize the generation on polluting residues without compromising the efficiency of the process [61,65].

## Figures and Tables

**Figure 1 ijms-26-03613-f001:**
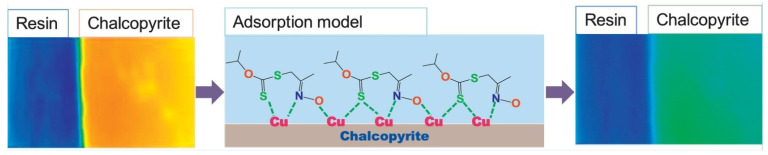
Adsorption model of IPXPO on the chalcopyrite surface.

**Figure 2 ijms-26-03613-f002:**
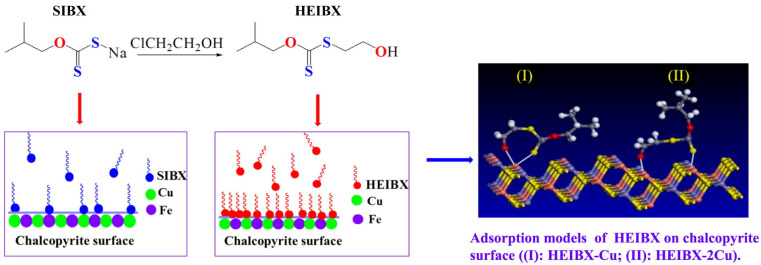
Adsorption model of HEIBX on the chalcopyrite surface. The formation of Cu–S and Cu–O bonds, which improve flotation selectivity, is observed [47].

**Figure 3 ijms-26-03613-f003:**
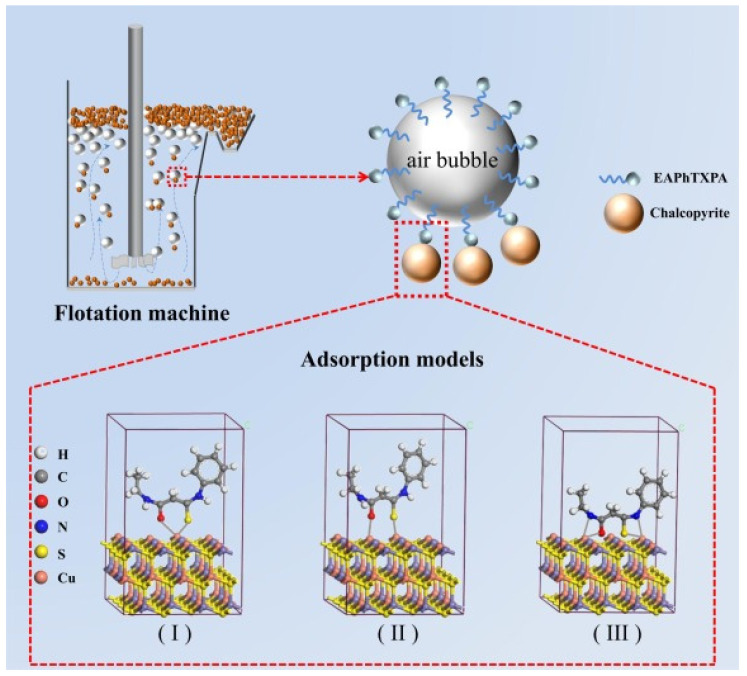
Adsorption model of EAPhTXPA on the chalcopyrite surface.

**Figure 4 ijms-26-03613-f004:**
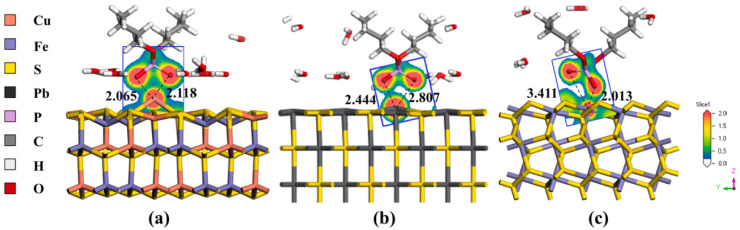
Adsorption models of HDBP on the surfaces of (**a**) chalcopyrite, (**b**) galena and (**c**) pyrite [60]. On chalcopyrite, HDBP forms stable Cu–O–P and Cu–O=P bonds, whereas no significant adsorption occurs on galena and pyrite.

**Figure 5 ijms-26-03613-f005:**
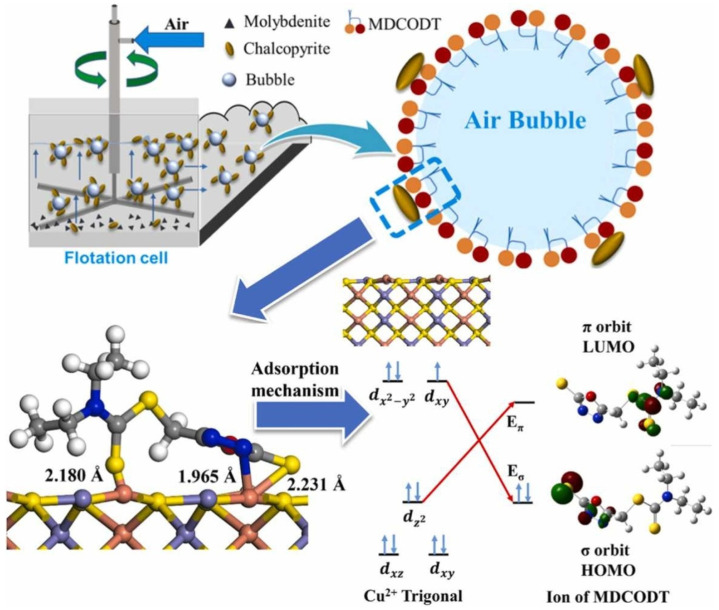
Selective adsorption mechanism of MDCODT on the chalcopyrite surface through Cu–S and Cu–N bonds that increase its hydrophobicity [63].

**Figure 6 ijms-26-03613-f006:**
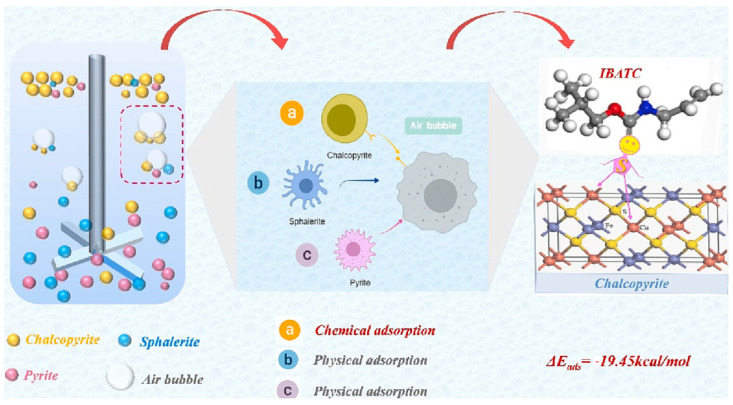
Adsorption model of IBATC on the chalcopyrite surface through Cu–S bonds which favor selective flotation [66].

**Figure 7 ijms-26-03613-f007:**
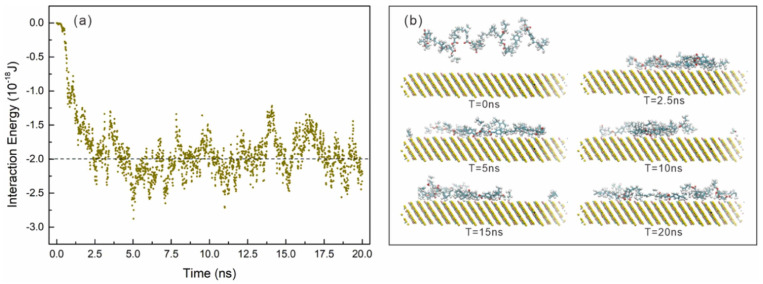
MD simulation on the interaction between the St–Ba molecule with the chalcopyrite surface [71]. (**a**) Interaction energy variation over time; (**b**) time-lapse capture of St–Ba disposition on the mineral surface.

**Figure 8 ijms-26-03613-f008:**
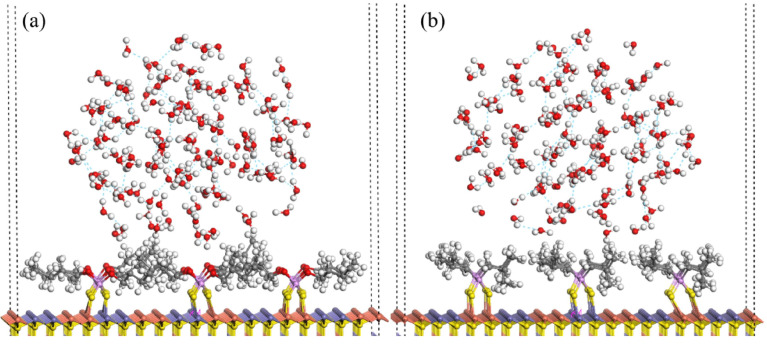
Hydrophobic structure of the (**a**) DTP and (**b**) 3418A collectors on chalcopyrite surface [81]. It is observed that 3418A forms a more compact and ordered coating, reducing its interaction with water and increasing mineral hydrophobicity.

**Figure 9 ijms-26-03613-f009:**
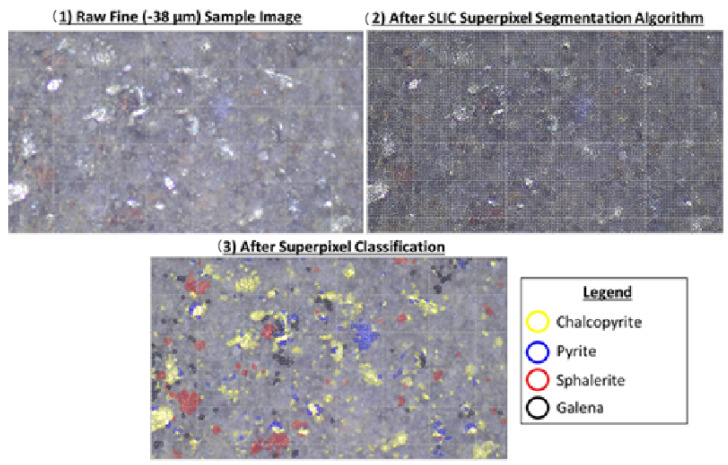
Segmentation and classification of sulfide mineral particles in flotation pulp. Observed are (1) original image of the fine fraction; (2) segmentation in super pixels; and (3) minerals classification [83].

**Figure 10 ijms-26-03613-f010:**
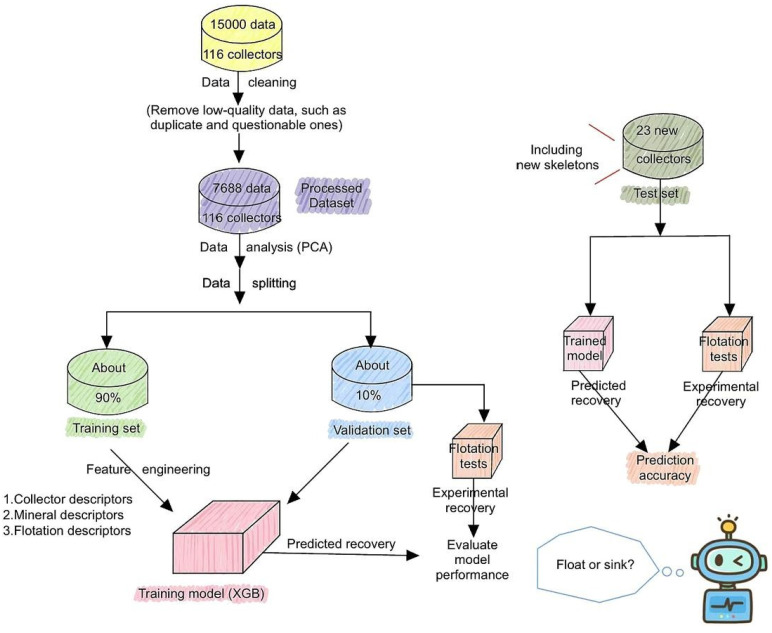
Workflow of the proposed model. The diagram shows the integration of QC with ML to characterize, predict and validate the performance of a group of collectors in the sulfide minerals flotation [22].

**Table 1 ijms-26-03613-t001:** Number of results per search query.

Search Query	Number of Results
Chalcopyrite flotation and copper recovery	81
Chalcopyrite flotation and reagent effect	104
Chalcopyrite flotation and surface characteristics	29
Chalcopyrite flotation and analytical techniques	37
Chalcopyrite flotation and molecular dynamics	19
Chalcopyrite flotation machine learning	8
Total results	278

**Table 2 ijms-26-03613-t002:** Papers classified by sections.

Section	Years	Number of Publications
First section	1979–2015	15
Second section	2009–2024	33
Third section	2018–2024	17

**Table 3 ijms-26-03613-t003:** Measurement techniques and their abbreviations.

Technique Name	Abbreviation
X-ray photoelectron spectroscopy	XPS
Fourier transform infrared spectroscopy	FTIR
Ultraviolet–visible spectroscopy	UV–Vis
Nuclear magnetic resonance spectroscopy	NMR
Local electrochemical impedance spectroscopy	LEIS
Atomic absorption spectroscopy	AAS
Raman spectroscopy	-
Secondary ion mass spectrometry	SIMS
Inductively coupled plasma mass spectrometry	ICP-MS
Atomic force microscopy	AFM
Scanning electron microscopy	SEM
Transmission electron microscopy	TEM
Scanning electrochemical microscopy	SECM
X-ray diffractometry	XRD

**Table 4 ijms-26-03613-t004:** Integrated summary of articles on analytical techniques and innovative reagents.

Reagent	Methods	Results	Reference
Collector: Butyl xanthate (BX). Depressants: Lignosulfonate calcium (LSC). Frother: 2# oil.	Separation of chalcopyrite and pyrite. Particle size < 106 µm. pH 9–10 fixed with NaOH. XFG-1600 flotation machine. FTIR to analyze LSC adsorption on mineral surfaces.	LSC selectively depressed pyrite without affecting chalcopyrite flotation. Optimal separation at pH 9–10 using 150 mg/L of LSC, 0.1 mmol/L of BX, and 15 mg/L of 2# oil. The copper concentrate grade reached 24.73% with a recovery of 80.36%. FTIR analysis confirmed that LSC adsorbed on pyrite, making it hydrophilic, whereas chalcopyrite was unaffected due to prior xanthate adsorption.	[36]
Collector: N/A. Depressant: Carboxymethyl cellulose (CMC) polymers with varying degrees of substitution. Frother: Methyl isobutyl carbinol (MIBC).	Separation of chalcopyrite and talc. Particle size < 850 µm. pH 9 fixed with HCl/KOH. Denver flotation cell. AFM to analyze adsorption. Contact angle to assess wettability. XPS and electron microprobe to confirm purity.	LSLB polymer reduced talc recovery (54%), while chalcopyrite recovery remained high (~91%). AFM and contact angle showed that CMCs with lower substitution and random distribution exhibited greater surface adsorption, reducing talc hydrophobicity and enhancing depressant performance.	[37]
Collector: Sodium butyl xanthate (SBX). Depressant: Sodium glycerine–xanthate (SGX). Frother: Pine oil.	Separation of chalcopyrite and pyrite. Particle size +38–74 µm. pH 7–10. microflotation cell. Zeta potential to assess SGX adsorption. UV–Vis to quantify SBX and SGX in solution. FTIR to identify adsorbed functional groups.	SGX effectively depressed pyrite (<10% recovery at 45 mg/L) while maintaining high chalcopyrite recovery (~80–90%). Zeta potential and UV–Vis confirmed stronger SGX adsorption on pyrite. FTIR showed SGX blocked active sites, preventing SBX adsorption and enabling efficient separation.	[38]
Collector: N,N–diethyl–N′–cyclohexylthiourea (DECHTU). Depressant: N/A. Frother: N/A.	Separation of chalcopyrite. Particle size +38−76 µm. pH 4–8. Hallimond cell. UV–Vis to quantify DECHTU in solution and calculate adsorption. Zeta potential to detect surface charge changes. FTIR and XPS to confirm chemisorption and surface complex formation.	DECHTU enabled effective chalcopyrite flotation (up to ~90% recovery) at pH 4–8. Adsorption followed Langmuir isotherm and pseudo–second–order kinetics. Thermodynamic parameters indicated spontaneous, exothermic chemisorption. Zeta potential and FTIR confirmed anionic adsorption. XPS verified Cu(I)–DECHTU complex formation.	[39]
Collector: 3–hexyl–4–amino–1,2,4–triazole–5–thione (HATT). Depressant: N/A. Frother: Methyl isobutyl carbinol (MIBC).	Chalcopyrite flotation. Particle size +37−74 µm. pH 3–10. Hallimond cell. UV–Vis to quantify HATT in solution. Zeta potential to assess surface adsorption. FTIR and XPS to confirm chemisorption and surface complex formation.	HATT enabled high chalcopyrite recovery (up to 96.07% at 2 × 10^−5^ mol/L) across pH 3–10, outperforming SIBX (88.77%). Adsorption followed Langmuir isotherm and pseudo–second–order kinetics. Thermodynamic data indicated spontaneous, endothermic chemisorption. Zeta and FTIR confirmed HATT–Cu complex formation.	[40]
Collector: Sodium butyl xanthate (SBX). Depressant: N/A. Frother: Terpenic oil.	Chalcopyrite flotation. Particle size +74–500 µm. pH 10. XFG flotation cell. Pulp potential measurement to assess Eh. XPS to analyze surface composition (<20 µm) and collector adsorption.	Adding SBX before grinding yielded higher recovery (93.62% vs. 90.03% at 2 × 10⁻³ mol/L) and higher pulp potential. XPS revealed higher collector adsorption (C: 20.11%), more free oxygen (O: 51.04%), and less oxidized Fe and Cu species in fine particles, improving flotation of <20 µm fraction.	[41]
Collector: O–isopropyl–S–[2–(hydroxyimino)propyl] dithiocarbonate ester (IPXPO). Depressant: N/A. Frother: Methyl isobutyl carbinol (MIBC).	Separation of chalcopyrite and pyrite. Particle size +38–76 µm. pH 4–9. Hallimond cell. UV–Vis to calculate adsorption. Zeta potential to assess surface interactions. In situ SECM to visualize surface coverage. FTIR to identify functional groups adsorbed.	IPXPO enabled high chalcopyrite recovery (~95%) and low pyrite recovery (~20%) at pH 4–9. Adsorption followed Langmuir isotherm and pseudo–second–order kinetics. Zeta, SECM and FTIR confirmed chemisorption of IPXPO, forming Cu–S, Cu–N and Cu–O bonds on the surface.	[42]
Collector: Nanoparticle emulsion. Depressant: N/A. Frother: N/A.	Separation of chalcopyrite, pyrite and quartz. Particle size < 23 µm. pH 2–10. Microflotation cell. FTIR to confirm chemical adsorption. Zeta potential to evaluate surface charge changes. SEM to observe morphology and adsorption.	HNP enabled high chalcopyrite recovery (up to 96.32% with 1 mL) at pH 6, selectively over pyrite (66.44%) and quartz (25.88%). FTIR confirmed chemical bonding between HNP and chalcopyrite. Zeta potential indicated anionic collector behavior. SEM showed strong adsorption on chalcopyrite and limited interaction with other minerals.	[43]
Collector: Butyl xanthate (BX). Depressant: Zinc sulfate (ZnSO_4_) and sodium dimethyl dithiocarbamate (SDD). Frother: N/A	Separation of chalcopyrite and sphalerite. Particle size +40–75 µm. pH 4–12. XFG microflotation cell. Zeta potential to evaluate adsorption. LEIS to map local electrochemical impedance and confirm surface coverage.	Optimal separation at pH 10 using 10^−4^ mol/L ZnSO_4_+SDD (3:1) and 10^−5^ mol/L BX. A concentrate containing 30.21% Cu with 86.79% recovery was obtained. Zn content was 4.20%, with a recovery of 5.48%. Zeta potential and LEIS confirmed that ZnSO_4_+SDD significantly prevented BX adsorption on sphalerite while allowing it on chalcopyrite.	[44]
Collector: Ammonium dibutyl dithiophosphate (ADD). Depressant: Sodium alginate (NaAl). Frother: Not specified.	Separation of chalcopyrite and galena. Particle size +74–106 µm. pH 7–12. XFG microflotation cell. Zeta potential to assess surface interactions. FTIR to identify adsorbed functional groups. XPS for surface analysis and adsorption confirmation.	NaAl reduced galena recovery (<10% at 15 mg/L), while maintaining chalcopyrite recovery (>80% across pH 7–12). In 1:1 mixed feed, NaAl+ADD yielded a concentrate with 65.69% chalcopyrite and 82.34% recovery. Zeta and FTIR confirmed strong NaAl chemisorption on galena, but not on chalcopyrite. XPS verified COO– groups and Pb signal changes, indicating selective surface coverage.	[45]
Collector: Butyl xanthate (BX). Depressant: Seaweed glue (SEG). Frother: Methyl isobutyl carbinol (MIBC).	Separation of chalcopyrite and galena. Particle size +38–74 µm. pH 8. Microflotation cell. Contact angle to assess wettability. UV–Vis to quantify adsorption. Zeta potential for surface interaction. FTIR to identify adsorbed functional groups.	SEG reduced galena recovery from 94.89% to 7.55% at 15 mg/L, while chalcopyrite recovery remained above 75%. In mixed ore (8.29% Cu), a concentrate with 23.68% Cu and 81.52% recovery was obtained. Analytical results confirmed strong chemisorption of SEG on galena and weak physisorption on chalcopyrite, enabling selective depression.	[46]
Collector: S–hydroxyethyl–O–isobutyl xanthate (HEIBX) compared with Sodium isobutyl xanthate (SIBX). Depressant: N/A. Frother: Methyl isobutyl carbinol (MIBC).	Separation of chalcopyrite and pyrite. Particle size +38−76 µm. pH 8. Microflotation cell. DFT to analyze molecular reactivity. UV–Vis to assess metal–ion interaction. Contact angle for wettability. Zeta potential, FTIR and XPS to confirm adsorption mechanisms.	HEIBX achieved 93.94% chalcopyrite recovery and 24.39% pyrite recovery (at 4 × 10^−5^ mol/L). DFT and UV–Vis confirmed selective interaction of C=S and OH groups with Cu⁺/Cu^2+^ forming Cu–O and Cu–S bonds. Zeta, FTIR and XPS showed strong chemisorption on chalcopyrite and weak interaction with pyrite, enabling efficient separation.	[47]
Collector: Thiohexanamide (THA) compared with O–isopropyl–N–ethyl thionocarbamate (IPETC) and sodium isobutyl xanthate (SIBX). Depressants: N/A. Frother: Methyl isobutyl carbinol (MIBC).	Separation of chalcopyrite, pyrite and galena. Particle size +38–76 µm. pH 8. Microflotation cell and bench-scale tests. UV–Vis to assess metal ion interaction. Zeta potential, FTIR and XPS to confirm adsorption mechanisms. DFT to explore electronic interactions with Cu.	THA achieved 97.1% chalcopyrite recovery (at 5 × 10^−5^ mol/L), while pyrite and galena were 34.8% and 3.5%, respectively. In mixed ore, the concentrate had 13.05% Cu and 94.22% recovery. UV–Vis confirmed selective THA–Cu⁺ interaction. Zeta, FTIR and XPS showed strong chemisorption on chalcopyrite and weak interaction with pyrite and galena. DFT supported Cu–THA complex formation via Cu–S and Cu–N bonding.	[48]
Collector: Sodium amyl xanthate (SAX). Depressant: Sodium alginate (NaAl). Frother: Methyl isobutyl carbinol (MIBC).	Separation of chalcopyrite, chlorite and serpentine. Particle size +38–75 µm. pH 9. Microflotation and batch cell. TOC to measure NaAl adsorption density. Zeta potential to track surface charge shifts. FTIR to confirm chemical adsorption.	NaAl reduced chlorite (60% to <5%) and serpentine (36% to <5%) recovery while maintaining chalcopyrite (~90%). In ternary mix, concentrate reached 31% Cu and 90% recovery. TOC showed selective adsorption on gangues (0.14–0.24 mg/m^2^), negligible on chalcopyrite (<0.02 mg/m^2^). FTIR and zeta confirmed selective chemisorption.	[49]
Collector: Potassium isobutyl xanthate (KIBX). Depressant: O–carboxymethyl chitosan (O–CMC). Frother: Methyl isobutyl carbinol (MIBC).	Separation of chalcopyrite and molybdenite. Particle size +74–150 µm. pH 3–11. Hallimond cell. AFM to observe surface adsorption of O–CMC on treated minerals.	O–CMC (150 ppm) strongly depressed molybdenite (<12% recovery) across pH 3–11, while chalcopyrite flotation remained largely unaffected (>93% recovery). AFM confirmed irreversible adsorption on molybdenite and negligible adsorption on chalcopyrite, enabling selective separation.	[50]
Collector: Sodium ethyl xanthate (SEX). Depressant: Native wheat starch (NWS) and oxidized starch (Ox 5/120). Frother: polyglycol ether mix (FZS180).	Separation of chalcopyrite and graphite. Particle size P80 = 212 µm. pH 7.5. Flotation cell. Adsorption to measure surface density. AFM to visualize polymer layer morphology on graphite.	Ox 5/120 (8 mg/L) depressed graphite (<42%) with limited effect on chalcopyrite (~79%). At higher dosage (20 mg/L), graphite recovery dropped more than chalcopyrite, indicating strong selectivity. AFM showed Ox 5/120 formed a well-structured network on graphite, increasing wettability. Lower adsorption on chalcopyrite (1.70 mg/m^2^) vs NWS (5.54 mg/m^2^) explained reduced depression.	[51]
Collector: Sodium isopropyl xanthate (SIPX), potassium amyl xanthate (PAX), thionocarbamate (TC). Depressant: Modified and commercial lignosulfonates (KLS, CLS). Frother: Methyl isobutyl carbinol (MIBC).	Separation of chalcopyrite and molybdenite. Particle size +75–150 and +38–75 µm, respectively. pH 6–9. Partridge–Smith cell. FTIR to identify functional groups. ICP–OES to measure Ca, Na. Electrophoretic mobility to assess surface interaction.	Lignosulfonates strongly depressed molybdenite, while PAX preserved chalcopyrite flotation (>90% recovery). SIPX led to strong depression of chalcopyrite. FTIR and mobility data confirmed interaction with surface Ca^2+^ sites on molybdenite. PAX hindered lignosulfonate adsorption on chalcopyrite. KLS were as effective as CLS.	[52]
Collector: 3–ethylamino–N–phenyl–3–thioxopropanamide (EAPhTXPA). Depressant: N/A. Frother: N/A.	Chalcopyrite flotation. Particle size +38–76 µm. pH 3–10. XFGCII flotation cell. Contact angle to assess hydrophobicity. UV–Vis to quantify residual collector. DFT for electronic reactivity. XPS to confirm Cu–S, Cu–O and Cu–N bonding.	EAPhTXPA achieved chalcopyrite recovery up to 97.5% (pH 8, 4 × 10^−5^ mol/L). Highest contact angle (97°), showing stronger collecting ability than EAMTXPA and PhATXPA. Adsorption was spontaneous, endothermic and chemisorptive. DFT identified C=S, C=O and NH as active sites. XPS confirmed Cu–EAPhTXPA complex formation.	[53]
Collector: 6–hexyl–1,2,4,5–tetrazinane–3–thione) (HTT) vs sodium hexyl xanthate (SHX). Depressant: N/A. Frother: Methyl isobutyl carbinol (MIBC).	Separation of chalcopyrite and pyrite. Particle size +38–76 µm. pH 6–11.5. Hallimond cell. AFM to observe surface coverage. Contact angle for hydrophobicity. UV–Vis for interaction with metal ions. DFT for electronic reactivity. Zeta potential for surface charge. XPS to confirm bonding.	HTT achieved selective flotation, with chalcopyrite (97.2%) and pyrite (22.4%) at pH 10.5 (2 × 10^−5^ mol/L). Contact angle rose to ~95° (vs. ~88.5° for SHX). AFM showed dense HTT adsorption. UV–Vis confirmed HTT prefers Cu(I/II) over Fe(II/III). DFT indicated higher electron-accepting ability. XPS confirmed Cu–S and Cu–N bonding on surface.	[54]
Collector: N–benzoyl–N’,N’–diethyl thiourea (BDETU) compared with O–isopropyl–N–etil tionocarbamato (IPETC). Depressant: N/A. Frother: Methyl isobutyl carbinol (MIBC).	Separation of chalcopyrite and pyrite. Particle size +38–76 µm. pH 2–11. Hallimond cell. Contact angle for hydrophobicity. UV–Vis for interaction with metal ions. FTIR to identify bonding groups. DFT to analyze active sites and binding energy.	BDETU achieved 98.0% chalcopyrite recovery (at pH 8, 2 × 10^−5^ mol/L), while pyrite recovery remained low (20.65%). Contact angle increased to 90.5° on chalcopyrite with minimal change on pyrite. UV–Vis and FTIR confirmed selective interaction via C=O and C=S groups, forming C–O–metal and C–S–metal bonds. DFT indicated six–membered chelate ring formation, enhancing adsorption.	[55]
Collector: 5–methyl isobutylxanthate–1,3,4–oxadiazole–2–thione (MIXODT) vs. Sodium Isobutyl Xanthate (SIBX). Depressant: N/A. Frother: Methyl isobutyl carbinol (MIBC).	Flotation of chalcopyrite. Particle size: +38–76 µm. pH 3–11. Microflotation cell. Contact angle for hydrophobicity. Zeta potential to assess surface adsorption. UV–Vis to detect interaction with metal ions. FTIR and XPS to confirm bonding. DFT to predict adsorption structures and binding energy.	MIXODT achieved 95% chalcopyrite recovery (1 × 10^−5^ mol/L, pH 3–9), outperforming 77% SIBX. Contact angle >80°, indicating enhanced hydrophobicity. Zeta potential showed anionic adsorption. UV–Vis, FTIR and XPS confirmed surface complex formation with Cu(I) via Cu–S and Cu–N bonds. DFT supported stable chemisorption models.	[56]
Collector: Ammonium dibutyl dithiophosphate (ADD). Depressant: Sodium polyaspartate (PASP). Frother: Methyl isobutyl carbinol (MIBC).	Separation of chalcopyrite and galena. Particle size +38–74 µm. pH 10. XFGCII cell. TOC to quantify PASP adsorption. UV–Vis to measure ADD adsorption. FTIR and XPS to analyze functional groups and surface bonding.	PASP effectively depressed galena (1.1% recovery), maintaining high chalcopyrite recovery (91.9%) with 10 mg/L PASP. TOC showed higher PASP adsorption on galena. UV–Vis confirmed that PASP reduced ADD adsorption on galena. FTIR and XPS verified chemical adsorption of PASP on both minerals, with stronger interaction on galena.	[57]
Collector: O–isopropyl–N–ethyl thionocarbamate (IPETC) + BEAT (O,O′–bis(2–butoxyethyl) ammonium dithiophosphate). Depressant: N/A. Frother: Methyl isobutyl carbinol (MIBC).	Separation of chalcopyrite and pyrite. Particle size +38–74 µm. pH 7.0–7.5. XFG cell. Foaming tests to assess stability. Contact angle for wettability. Surface tension to evaluate solubility. FTIR to analyze adsorption on chalcopyrite. Intermolecular interaction parameter (β) calculation.	The IPETC/BEAT mixture (60% molar BEAT) achieved highest chalcopyrite recovery (92.43%) and low pyrite recovery (~14.2%) at 4 × 10^−5^ mol/L. Contact angle on chalcopyrite rose to 117.7°, indicating improved hydrophobicity. Lower surface tension and CMC values suggested enhanced solubility. FTIR confirmed co–adsorption on chalcopyrite. Negative β (−3.05) confirmed synergistic effect.	[58]
Collector: Sodium isoamyl xanthate (SIX). Depressant: Tea polyphenols (TP). Frother: Methyl isobutyl carbinol (MIBC).	Separation of chalcopyrite and galena. Particle size: 45–75 µm. pH 8. XFGCII cell. Contact angle for hydrophobicity. Zeta potential to study surface interactions. UV–Vis to quantify unadsorbed collector. ToF–SIMS to detect adsorbed ionic species. XPS to analyze surface chemical changes.	TP reduced galena recovery to 6.01%, while chalcopyrite recovery remained at 85.63% (TP 4 × 10^−4^ mol/L, SIX 2 × 10^−4^ mol/L). UV–Vis showed that TP inhibited SIX adsorption on galena. Contact angle and zeta results confirmed stronger hydrophobicity on chalcopyrite and preferential TP adsorption on galena. XPS and ToF-SIMS revealed Pb–TP complex formation on galena, blocking active sites.	[59]
Collector: Dibutyl phosphonate (HDBP). Depressant: N/A. Frother: Terpineol.	Separation of chalcopyrite, galena and pyrite. Particle size +37–74 µm. pH 6–8. XFG flotation cell. XPS to analyze elemental composition and surface bonding. CV and Tafel for electrochemical reactivity. DFT to simulate adsorption energy and bonding on mineral surfaces.	HDBP enabled selective flotation of chalcopyrite (>85% recovery at 5 × 10^−5^ mol/L) over galena (≤19%) and pyrite (≤8%). XPS showed P–O and P=O bonding only on chalcopyrite. Electrochemical tests confirmed strong reactivity with chalcopyrite but minimal changes for galena and pyrite. DFT showed highest adsorption energy on chalcopyrite, forming stable Cu–O–P and Cu–O=P chelation bonds.	[60]
Collector: Ethylenediamine tetramethylenephosphonic acid (EDTMPA). Depressant: N/A. Frother: Terpineol.	Separation of chalcopyrite and pyrite. Particle size +35−74 µm. pH 6–11. XFG cell. XPS to analyze surface bonding. FTIR to identify adsorbed functional groups. CV and Tafel for electrochemical reactivity. DFT and crystal chemistry to evaluate adsorption energy and surface affinity.	EDTMPA achieved high chalcopyrite recovery (~88.36%) and low pyrite recovery (~13.93%) at pH 9.0 (1 × 10⁻⁴ mol/L). XPS and FTIR showed strong binding of P=O and P–O groups to Cu/Fe sites on chalcopyrite. DFT revealed higher adsorption energy on chalcopyrite. Crystal chemistry confirmed greater metal atom density and valence.	[61]
Collector: 3–pentadecylphenyl 4–(3,3–diethylthiouredo–4–oxobutanoate) (cardanol derivative DP089). Depressant: N/A. Frother: Methyl isobutyl carbinol (MIBC).	Separation of chalcopyrite and pyrite. Particle size +53–104 µm. pH 8. Denver cell. UV–Vis to assess metal ions affinity. FTIR to identify bonding. Contact angle for hydrophobicity. Adsorption isotherms for kinetic modeling.	DP089 showed strong selectivity for chalcopyrite (92.1% recovery) and low recovery of pyrite (11.2%) at pH 8. In mixed mineral flotation, 87.3% chalcopyrite recovery and 12.7% pyrite recovery were achieved. UV–Vis confirmed selective interaction with Cu⁺. FTIR revealed formation of C–O–Cu and C–S–Cu bonds. Contact angle increased to 97.3° in chalcopyrite, confirming enhanced hydrophobicity.	[62]
Collector: 5–methyl diethyl dithiocarbamate–1,3,4–oxadiazole–2–thione (MDCODT). Depressant: N/A. Frother: Not specified.	Separation of chalcopyrite and molybdenite. pH 9. XFG–II flotation cell. Zeta potential to assess surface charge. UV–Vis to analyze adsorption behavior. Contact angle for hydrophobicity. AFM to examine surface morphology. FTIR and XPS to confirm bonding. Electrochemistry and DFT to evaluate interaction and adsorption mechanism.	MDCODT achieved selective flotation of chalcopyrite (~95% recovery) and low molybdenite recovery (<20%) at 1 × 10⁻⁵ mol/L and pH 9. Zeta potential and contact angle (76°) confirmed selective adsorption. UV–Vis, FTIR and XPS verified Cu–S and Cu–N bonding. AFM and electrochemical tests showed dense surface adsorption. DFT supported stable chemisorption via Cu complexation.	[63]
Collector: Potassium ethyl xanthate (KEX). Depressant: Calcium lignosulfonate (CLS). Frother: Methyl isobutyl carbinol (MIBC).	Separation of chalcopyrite and molybdenite. Particle size <38 µm. pH 8. XFG cell. FTIR for chemical adsorption. XPS to identify surface bonding. UV–Vis to quantify CLS adsorption/desorption.	CLS depressed molybdenite flotation (~18% recovery at 50 mg/L) with minimal effect on chalcopyrite (~80%). In mixed minerals, Cu recovery was 84% and Mo 35%, with grades of 29.56% Cu and 5.17% Mo. FTIR and XPS confirmed chemisorption on chalcopyrite and hydrophobic interaction on molybdenite. UV–Vis showed higher CLS desorption from chalcopyrite than from molybdenite.	[64]
Collector: Poly(CA4–co–ACOEA14 (RAFT) with O–ethyl acetylcarbamothioate functionality. Depressant: N/A. Frother: N/A.	Separation and flocculation of chalcopyrite and pyrite. Particle size: 53–104 µm (flotation). pH 8. Contact angle to assess hydrophobicity. UV–Vis for adsorption quantification. FTIR and XPS to confirm chemical interaction.	In mixed mineral, poly(CA4–co–ACOEA14) achieved 82.3% chalcopyrite and 17.6% pyrite recovery (1 × 10^−5^ mol/L). FTIR and XPS confirmed selective chemisorption on chalcopyrite. Higher contact angle indicated greater hydrophobicity.	[65]
Collector: O–isobutyl–N–allyl thionocarbamate (IBATC). Depressant: N/A. Frother: Not specified.	Separation of chalcopyrite from pyrite and sphalerite. Particle size −38 µm. pH 9.5–10. XFG cell. FTIR and XPS to confirm chemisorption. UV–Vis to quantify adsorption. Zeta potential for surface interactions. DFT to calculate adsorption energy.	IBATC achieved high chalcopyrite recovery (>80%) at pH 9.5–10, with low sphalerite (<40%) and pyrite (~20%) recovery. Adsorption was 81–128% higher on chalcopyrite. FTIR and XPS confirmed chemisorption of sulfur atom onto Cu.	[66]
Collector: O–methylisobutyl-N–allyl thionocarbamate (MIBATC). Depressant: N/A. Frother: Terpineol.	Separation of chalcopyrite, sphalerite and pyrite. Particle size: −38 μm. pH 2–12. XFG flotation cell. FTIR and XPS to analyze adsorption and active sites. Zeta potential for surface change. UV–Vis to determine adsorbed amount. DFT to calculate adsorption energy and orbital interaction.	Chalcopyrite recovery > 80% at pH 9.5–10 with 30–40 mg/L MIBATC. Adsorption on chalcopyrite was 167% and 125% higher than on sphalerite and pyrite. Zeta potential showed positive shift. FTIR and XPS confirmed chemisorption via Cu–S bond. DFT indicated spontaneous, exothermic reaction.	[67]
Collector: 5–methyl butylxanthate–1,2,4–triazole–3–thione (MBuXTT). Depressant: N/A.Frother: N/A.	Separation of chalcopyrite and pyrite. Particle size +38–74 µm. XFG-II flotation cell. Contact angle to assess hydrophobicity. UV–Vis to study interaction with metal ions. Zeta potential to assess surface affinity. FTIR and XPS to identify adsorbed groups. DFT to confirm bonding behavior.	MBuXTT achieved 96% chalcopyrite recovery at pH 7 with 3 × 10^−5^ mol/L, with 23% pyrite recovery. Contact angle reached 89.5° (vs 83.5° for SBX). Zeta potential indicated higher chalcopyrite affinity. FTIR and XPS confirmed Cu–S and Cu–N bonding. DFT supported stable surface complex formation.	[68]

**Table 5 ijms-26-03613-t005:** Software for molecular dynamic simulation.

Software Name	Reference
Materials Studio 2020	[69,70,71,72,73,74,75,76,77,78]
LAMMPS 2020	[71]
Gaussian 16	[22,79,80]
DBTF+	[81]
CP2K 2020	[78]

**Table 6 ijms-26-03613-t006:** Integrated summary of articles on molecular design.

Reagent	Methods	Results	Reference
Collector: O-isopropyl-N-ethyl thionocarbamate (IPETC). Depressant: N/A. Frother: N/A.	Separation of chalcopyrite and galena. MDS in Materials Studio. PCFF force field. Chalcopyrite (112) and galena (110) surface. Geometry optimized with CASTEP and DMol^3^. Adsorption energy via Discover. NVT at 298.15 K, 2 ns, Nose thermostat. Ewald summation for interactions. Flotation −74 µm, pH 4–13, XFD cell. FTIR, UV–Vis to evaluate adsorption.	MDS showed stronger interaction of IPETC with galena (−47.39 kJ/mol) than with chalcopyrite (−29.73 kJ/mol). However, IPETC displaced H_2_O and OH⁻ on chalcopyrite, allowing competitive adsorption. At pH 9.5 and 7 × 10^−4^ mol/L IPETC, chalcopyrite recovery reached 83.6%, while galena was 61.8%, achieving effective separation. FTIR confirmed adsorption of –C=N and –CH groups on both minerals at pH 9.5.	[69]
Collector: Butyl xanthate (BX). Depressant: Peracetic acid (PAA). Frother: Not specified.	Separation of chalcopyrite and arsenopyrite. MDS in Materials Studio (Forcite). Universal force field. NVT at 298 K for 100 ps with Nose thermostat. Chalcopyrite (112) and arsenopyrite (001) surfaces modeled as 4 × 3 × 1 supercells with 15 Å vacuum. Adsorption energy and electronic interaction via DFT using CASTEP, GGA/PW91 functional, 400 eV cutoff, k–points grids 2 × 2 × 1 and 3 × 4 × 1. Verified with XPS.	PAA adsorbed more strongly on arsenopyrite (−550.5 kJ/mol) than chalcopyrite (−60.3 and −36.2 kJ/mol for Cu and Fe). MDS showed diffusion on arsenopyrite and aggregation on chalcopyrite, confirming preference. DOS and Mulliken analysis revealed O–Fe hybridization and spin polarization. XPS confirmed Fe(II) oxidation to Fe(III), preventing BX adsorption on arsenopyrite.	[70]
Collector: Styrene–butyl acrylate (St-Ba). Depressant: N/A. Frother: Methyl isobutyl carbinol (MIBC).	Separation of chalcopyrite and serpentine. MDS in LAMMPS. CVFF force field. Chalcopyrite (112) surface and St–Ba polymer model (5 monomers). Periodic box, NVT ensemble at 300 K, 0.1 MPa, Lennard–Jones potential, 1 fs timestep, 1000 outputs every 20 ps. Microflotation at −38 µm, pH 7, XFGC II cell. Contact angle, FTIR, SEM to evaluate adsorption.	St–Ba–chalcopyrite interaction energy was −1.99 × 10^−18^ J, indicating spontaneous adsorption. St–Ba adhered and spread on chalcopyrite at 2.5 ns. Chalcopyrite recovery increased from 89% (SBX) to 95% (St–Ba) for single mineral, and from 42% to 84% in mixture. No St–Ba adsorption on serpentine. Contact angle > 70° for chalcopyrite with St–Ba. FTIR and SEM confirmed effective adsorption.	[71]
Collector: Butyl xanthate (BX). Depressant: Sodium thioglycollate (STG). Frother: Methyl isobutyl carbinol (MIBC).	Separation of chalcopyrite and arsenopyrite. MDS with Materials Studio (Forcite). Universal force field. NVT at 298 K, 500 ps, 1 fs timestep. DFT using CASTEP, GGA/PW91 functional, 400 eV cutoff, k-points 2 × 2 × 1. Arsenopyrite (001) surface. Flotation at −74 + 38 µm, pH 8, 1 × 10^−5^ mol/L BX, 20 mg/L MIBC. LEIS, contact angle, FTIR, UV–Vis for adsorption analysis.	STG formed S–Fe and S–As bonds (up to −223.34 kJ/mol) and hydrogen bonds with water via –COO⁻. MDS confirmed stable adsorption (−212.55 kJ/mol). STG sharply reduced arsenopyrite recovery to ~5% while chalcopyrite remained ~90% at optimal dosage (1.25 × 10^−4^ mol/L). LEIS and FTIR validated selective adsorption. Contact angle difference (15.12°) confirmed hydrophilic film formation on arsenopyrite.	[72]
Collector: O–isobutil–N–hidroxietil tionocarbamato (IBHETC). Depressant: N/A. Frother: Methyl isobutyl carbinol (MIBC).	Separation of chalcopyrite and pyrite. IBHETC was synthesized and confirmed by GC, NMR, MS, FTIR. MDS were performed to study its interaction with Cu⁺. Initial structures were generated with GFN0–xTB (xtb 6.4.1), selected using Molclus, optimized by DFT in Gaussian 16, B3LYP-D3(BJ) functional, SDD basis for Cu, 6–311+G(d,p) for other, with IEF–PCM solvation. Flotation at −74 + 38 µm, pH 8, XFG–II cell. Adsorption analysis by FTIR, XPS.	MDS revealed that IBHETC forms a Cu⁺ complex via N–Cu (2.18 Å), O–Cu (2.13 Å), and S–Cu (2.31 Å) bonds, stabilized by four and five membered rings. The molecule undergoes deprotonation and isomerization, enhancing selectivity. Chalcopyrite recovery was 94.41% (vs. 81.79% with IBETC), while pyrite recovery was 12.63%. FTIR and XPS confirmed selective chemisorption on chalcopyrite.	[79]
Collector: Dibutyl dithiophosphate (DTP) and sodium diisobutyl dithiophosphinate (3418A). Depressant: Not specified. Frother: Not specified.	Flotation of chalcopyrite. Microflotation at −74 + 38 µm, pH 3–11, XFG5–35 cell. Microcalorimetry at 26 °C, 1×10⁻⁴ mol/L. UV–Vis for adsorption. CV for redox behavior. DFT using CASTEP, GGA–PW91 functional, 400 eV cutoff, 3 × 4 × 1 k–points. MDS with DFTB+. NVT at 298 K, 0.6 ps, 1 fs timestep.	3418A showed higher chalcopyrite recovery (>80%) and greater adsorption than DTP. Adsorption heats of 3418A were higher. MD showed stronger hydrophobicity (RDF peak 8.2 Å vs. 6.5 Å). DFT revealed higher adsorption energy (−311.4 vs. −170.9 kJ/mol), shorter Cu–S bonds, and stronger Mulliken populations. HOMO of 3418A (−4.098 eV) closer to chalcopyrite LUMO (−3.8 eV).	[81]
Collector: Sodium butyl xanthate (SBX). Depressants: Oxalic acid (OA) and Carboxymethyl cellulose (CMC). Frother: Methyl isobutyl carbinol (MIBC).	Separation of chalcopyrite and talc. Microflotation at −74 + 38 µm, pH 9, XFG–II cell. Bench–scale flotation (Cu 0.60%). Contact angle, Zeta potential, AFM, SEM–EDS, FTIR, TOC used to analyze surface properties and surface interactions. MDS in Materials Studio (CASTEP, Amorphous Cell, Forcite). Compass II force field. NVT at 298 K, 200 ps. Talc (001) surface.	OA+CMC (60 + 60 mg/L) achieved 86.54% chalcopyrite and 10.16% talc recovery (pH 9). Contact angle and zeta potential showed enhanced surface hydrophobicity difference. MDS revealed OA enables multilayer CMC adsorption on talc, introducing hydrophilic groups. In Bench–scale test, OA+CMC (30 + 30 g/t) gave Cu recovery of 85.85% with MgO recovery of 10.23%.	[73]
Collector: Sodium butyl xanthate (SBX). Depressant: Pyrogallic acid (PA). Frother: Terpineol.	Separation of chalcopyrite and galena. Microflotation at −200 + 75 µm, pH 9, synthetic resin cell. Contact angle, FTIR, TOC, UV–Vis, ToF–SIMS and XPS to assess adsorption and surface properties. MDS in Materials Studio (Forcite). Universal force field. NVT at 298 K, 1 fs, 1000 ps. Galena (100) surface.	PA (2 × 10^−4^ mol/L) enabled 83.56% chalcopyrite and only 11.22% galena recovery. MDS showed PA’s affinity to galena, promoting H_2_O accumulation and reducing hydrophobicity. Contact angle, FTIR, TOC, XPS and ToF–SIMS confirmed strong PA adsorption on galena, suppressing SBX adsorption.	[74]
Collector: Sodium butyl xanthate (SBX). Depressant: Mercaptosuccinic acid (MSA). Frother: Terpineol.	Separation of chalcopyrite and galena. Microflotation at pH 9. Contact angle to assess wettability. FTIR, UV–Vis, TOC, XPS, AFM to analyze adsorption and surface properties. MDS in Materials Studio (CASTEP, Amorphous Cell, Forcite). Universal force field. NVT at 298 K, 1000 ps. Galena (100) surface.	MSA reduced galena recovery to 16.12% at 8 × 10^−4^ M, while chalcopyrite remained > 80%. Contact angle, TOC, FTIR and XPS confirmed higher MSA and lower SBX adsorption on galena. MDS showed MSA increased H_2_O accumulation on galena, enhancing hydrophilicity and depression.	[18]
Collector: Sodium butyl xanthate (SBX). Depressant: Polymaleic acid (PMA). Frother: Terpineol.	Separation of chalcopyrite and galena. Microflotation at −74 + 38 µm, pH 9. Contact angle, FTIR, XPS, ToF–SIMS, UV–Vis, TOC to assess adsorption and surface properties. MDS in Materials Studio (Forcite). Universal force field. NVT 298 K, 1 fs, 1000 ps, 2000 H_2_O ± 4 PMA.	PMA (2 × 10^−4^ mol/L) depressed galena (−74.91%) but preserved chalcopyrite recovery (83.27%). In binary flotation, PMA improved Cu grade/recovery (28.11%/86.01%) and reduced Pb floatability. XPS, ToF–SIMS and MDS showed PMA increased hydrophilicity and water/PMA accumulation on galena.	[75]
Collector: Sodium butyl xanthate (SBX). Depressant: Carboxymethyl–β–cyclodextrin (CMCD). Frother: Pine oil.	Separation of chalcopyrite and pyrite. Microflotation at −74 + 38 µm, pH 6.5 ± 0.2, XFG cell. Contact angle, UV–Vis, FTIR, XPS, and AFM to assess surface interactions. MDS using Materials Studio (CASTEP, Amorphous Cell, Forcite). Universal force field. NVT at 98 K, 500 ps, 1 fs.	CMCD showed stronger affinity for pyrite (−430.10 kcal/mol) than for chalcopyrite (−179.40 kcal/mol). At 250 mg/L CMCD and 40 mg/L SBX, Cu recovery reached 73.15% with a grade of 28.06%. FTIR, XPS and AFM confirmed selective adsorption on pyrite via hydrogen bonding, preventing SBX adsorption.	[76]
Collector: PBX (Potassium n-butyl xanthate). Depressant: Tannic acid (TA). Frother: Methyl isobutyl carbinol (MIBC).	Separation of chalcopyrite and molybdenite. Microflotation at −76 + 38 µm, pH 10. UV, FTIR and XPS to analyze adsorption. MDS in Materials Studio (DMol^3^, CASTEP, Forcite). Universal force field. NVT (1 ps), NVE (50 ps), final NVT (1 ns, 1 fs timestep). Chalcopyrite (012) and molybdenite (001) surfaces.	TA enabled selective depression of molybdenite. Chalcopyrite recovery reached 95.01% and molybdenite 3.34% at 900 mg/L TA. XPS showed chemisorption on chalcopyrite via Fe, not Cu, and no chemisorption on molybdenite. MDS confirmed hydrophobic interaction as the main adsorption mechanism on molybdenite (−76.49 kcal/mol vs. −6.84 on chalcopyrite).	[77]
Collector: Sodium isobutyl xanthate (SIBX) and Kerosene. Depressant: Sodium thioglycolate (STG). Frother: Pine oil.	Separation of chalcopyrite and molybdenite. Microflotation at −74 + 38 μm, pH 10, XFGCII cell. Contact angle, UV, FTIR, XPS, ToF–SIMS, OCP, EIS to evaluate surface adsorption and hydrophobicity. MDS and AIMD with Materials Studio (CASTEP, Forcite, CP2K). Chalcopyrite (012) surface. COMPASS II force field. 4000 H_2_O, 40 STG^−^, 40 Na⁺. PBE–D3BJ functional. 298 K, 1 fs, 500 ps (MD), 5000 fs (AIMD).	STG enabled effective separation with 6.69% chalcopyrite and 55.26% molybdenite recovery, achieving 83.79% separation efficiency. XPS confirmed chemisorption via Cu(I)–S and Fe(III)–S bonds. Adsorption on chalcopyrite was ~4× higher than on molybdenite. MDS and AIMD showed multilayer STG adsorption, stronger water affinity, and adsorption energy of −57.48 kcal/mol. STG also removed 81% of SIBX from the chalcopyrite surface.	[78]

**Table 7 ijms-26-03613-t007:** Integrated summary of articles on machine learning.

Reagent	Methods	Results	Reference
Collector: Sodium isopropyl xanthate (SIPX). Depressant: Sodium cyanide (NaCN) and zinc sulfate (ZnSO_4_). Frother: Methyl isobutyl carbinol (MIBC).	Flotation of chalcopyrite and galena. Box–Behnken Design (BBD) with 8 variables (reagent dosages, pH, airflow, impeller speed, time). A hybrid ML model (RF–FFA) is applied to predict Cu/Pb grades and recoveries, and to compare its performance with other models (ANN, SVM, M5P, RF). The model is trained and validated using the BBD dataset. CPI analysis is conducted to identify the most influential parameters.	The RF–FFA hybrid model outperformed all standalone models. It achieved R^2^ values up to 0.9864 (Pb grade) and 0.9817 (Cu recovery), with minimal RMSE, MAE, and MAPE. CPI identifies SIPX and NaCN as the most influential reagents. Accurate predictions of flotation performance using process parameters enable efficient process optimization, reducing experimental cost and time.	[82]
47 ethyl–linked molecules with donor atoms (O, N, S).	QC+ML model to screen flotation reagents for chalcopyrite. DFT using B3LYP/def2–TZVP, SMD solvation and D3 dispersion was used to calculate ΔG and descriptors (15 total). Gaussian was used for QC. ML models included RR, RF, GBR and MLP. SBI to assess selectivity.	GBR was the most accurate model (R^2^ 0.90, RMSE 4.04 for Cu⁺). QC showed higher affinity for Cu(II), moderate for Fe(II) and lower for Cu(I). ML accurately predicted ΔG and SBI, identifying 8 reagents with highest selectivity. HOMO, LUMO and molecular charge were key descriptors. ML reduced screening time from ~38.5 h (QC) to ms. Cu–selective reagents mainly contained sulfur.	[80]
Collector: Isopropyl ethyl thionocarbamate (IET) and Sodium diisobutyl dithiophosphate (SDD). Depressant: Sodium metabisulfite (SMBS), zinc sulfate. Frother: Methyl isobutyl carbinol (MIBC).	DL model using SOLOv2 for instance segmentation and SLIC for unsupervised clustering to classify chalcopyrite, pyrite, sphalerite, galena and gangue in flotation pulp images. SOLOv2 trained in PyTorch for coarse (+38 µm), SLIC superpixels applied to fine (−38 µm). Validation via ICP–MS and XRD.	Vision model predicted mineral grades and P50 with high accuracy (±5.4% and ±2.3 µm). Repeated analysis reduced error to ±1.3%. ICP–MS and XRPD confirmed precise classification. Enabled sulfide identification in pulp in under 5 min, improving real–time flotation monitoring.	[83]
Collector: 139 flotation collectors with diverse molecular structures. Depressant: Not specified. Frother: Not specified.	QC+ML model to predict flotation recovery of chalcopyrite, galena, pyrite and sphalerite. Descriptors from DFT (Gaussian, MN15/def2–TZVP, Multiwfn) and mineral features from CP2K (PBE–D3, 400 Ry). ML model (XGBoost) was trained on 7688 tests using 69 input features (collector, mineral, flotation conditions).	The model accurately predicted flotation performance (MAE = 10% in validation, 5.2% in test). Enabled structure–performance insights. Effective for chalcopyrite, pyrite, galena and sphalerite. New selective reagents identified. Demonstrated potential for high-throughput screening.	[22]

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
