# Peer review of "Chalcopyrite Flotation, Molecular Design and Smart Industry: A Review"

_ijms, 2025, doi:10.3390/ijms26083613_

Round 1
Reviewer 1 Report
Comments and Suggestions for Authors
- The abstract does not clearly describe the methodology used for literature selection and classification, which makes it difficult to assess the comprehensiveness of the review.
- The introduction does not explicitly state the novelty of this review compared to existing literature on chalcopyrite flotation.
- The review primarily summarizes previous studies but lacks comparative analysis to highlight key differences in flotation techniques, reagent efficiency, and experimental conditions. Suggestion: Include a comparative table summarizing reagent types, flotation conditions, and recovery rates to enhance clarity and facilitate critical discussion.
- While molecular dynamics simulations are discussed, the manuscript lacks details on computational parameters such as force fields, simulation time, and validation methods. Provide specifics on MD simulation conditions and include a comparative discussion of different computational approaches used in flotation research.
- The section on smart industry and machine learning lacks concrete examples of how ML models have been successfully applied in flotation processes.
- Different studies in the review use varying experimental conditions, but no discussion is provided on the impact of these variations on flotation performance.
The English could be improved to more clearly express the research.
Author Response
Replies to Reviewer 1:
Comment 1: The abstract does not clearly describe the methodology used for literature selection and classification, which makes it difficult to assess the comprehensiveness of the review.
Response 1: Thank you very much for your feedback. This has been improved with lines 16 and 17
Comment 2: The introduction does not explicitly state the novelty of this review compared to existing literature on chalcopyrite flotation.
Response 2: Thank you very much for the comment. A paragraph has been added from lines 82 to 86.
Comment 3:The review primarily summarizes previous studies but lacks comparative analysis to highlight key differences in flotation techniques, reagent efficiency, and experimental conditions. Suggestion: Include a comparative table summarizing reagent types, flotation conditions, and recovery rates to enhance clarity and facilitate critical discussion.
Response 3: Table 4 has been included in the article.
Comment 4: While molecular dynamics simulations are discussed, the manuscript lacks details on computational parameters such as force fields, simulation time, and validation methods. Provide specifics on MD simulation conditions and include a comparative discussion of different computational approaches used in flotation research.
Response 4: Table 5 has been included in the article.
Comment 5: The section on smart industry and machine learning lacks concrete examples of how ML models have been successfully applied in flotation processes.
Response 5: Table 6 has been included in the article.
Comment 6: Different studies in the review use varying experimental conditions, but no discussion is provided on the impact of these variations on flotation performance.
Response 6: The discussion section has been added to the article.
Reviewer 2 Report
Comments and Suggestions for Authors
I suggest to mention in the introduction the need for copper out of the change to renewable energy (PV, windmills, grid) as a demand source. please avoid quotation lumps like [8-11] and give a short phrase per source. Can you introduce some structure into 3.1. Can you maybe start with a table, giving the key aspects of the papers you then discuss?
in figure 1, separate the caption from the explanation (text). same in figure 3. can you include in the description of papers also a brief assessment, what of the work described has found industrial application? Any observations regarding cost/benefit of flotation agents?
Please spellcheck thoroughly, also punctuation marks, use of capitals. demonstrated through
Author Response
Replies to Reviewer 2:
I suggest to mention in the introduction the need for copper out of the change to renewable energy (PV, windmills, grid) as a demand source.
Response: Thank you very much for your feedback. This has been included lines 36 to 42.
Can you maybe start with a table, giving the key aspects of the papers you then discuss?
Response: Table 4 has been included in the article.
In figure 1, separate the caption from the explanation (text). same in figure 3.
Response: Done
an you include in the description of papers also a brief assessment, what of the work described has found industrial application?
Response: Has been included lines 849 to 853 in the section discussion
Any observations regarding cost/benefit of flotation agents?
Response: Has been included lines 839 to 846 in the section discussion